# Evidence of an allostatic response by intestinal tissues following induction of joint inflammation

**Meghan M. Moran** [1,2,3]*, **Jun Li**[3], **Quan Shen**[4], **Sheona P. Drummond**[5], **Caroline M. Milner**[5], **Anthony J. Day** [5], **Ankur Naqib**[1,6], **D. Rick Sumner**[1,2], **Anna Plaas**[3]

**1** Department of Anatomy & Cell Biology, Rush University Medical Center (RUMC), Chicago, Illinois, United States of America, **2** Department of Orthopedic Surgery, RUMC, Chicago, Illinois, United States of America, **3** Department of Int. Medicine (Rheumatology), RUMC, Chicago, Illinois, United States of America, **4** Department of Neurosurgery, RUMC, Chicago, Illinois, United States of America, **5** Manchester Cell-Matrix Centre & Lydia Becker Institute of Immunology and Inflammation, Faculty of Biology, Medicine and Health, The University of Manchester, Manchester, United Kingdom, **6** Bioinformatics Core, RUMC, Chicago, Illinois, United States of America

* meghan_moran@rush.edu

## Abstract

Disrupted intestinal epithelial barrier function has been proposed to be integral to rheumatoid arthritis (RA) progression and pathogenesis. To further define the molecular pathways in synovial inflammation and the response of the intestinal tissues, we have used a rat model of mono-joint inflammatory arthritis, induced by intra-articular injection of Complete Freund's adjuvant (CFA). The predominant inflammatory response of a single injection of the adjuvant into the knee joint resulted in rapid and reproducible formation of a fibrotic myeloid-infiltrated synovial pannus. Our aim was to determine how intestinal tissues, including the proximal and distal ileum and distal colon, responded to inflammatory changes in the synovium in a temporally coordinated manner by comparing their transcriptomic landscapes using RNASeq analyses. We confirmed the timeline of joint inflammation by knee joint swelling measurement, increased synovial fluid levels of bikunin (a component of both the acute phase protein pre-alpha-inhibitor and inter-alpha-inhibitor) and demonstrated a self-correcting response of trabecular and cortical bone to the CFA challenge. Intestine-specific responses were monitored by 16S microbiome amplicon sequencing, histopathology for mucus layer integrity, and immune cell immunohistochemistry. We present data that shows the intestinal tissue displays an allostatic response to the acute joint inflammation and was region specific. The ileum primarily responded with increased mucus secretion and silencing of T-cell specific pathways, whereas the colon showed a transient upregulation of macrophages, with a broader suppression of immune related and metabolic pathway related transcripts. Interestingly, many neuropathways were activated early but then suppressed later in both the ileum and colon. There were only insignificant changes in the fecal microbiome composition in ileum or colon post-CFA administration. In summary, our data show for the first time

**Data availability statement:** "Yes - all data are fully available without restriction; All data can be found within this paper and its Supporting Information files. All sequencing reads generated in this study have been deposited in the National Center for Biotechnology Information (NCBI) BioProject database [https://www.ncbi.nlm.nih.gov/bioproject?db=bioproject] under accession numbers PRJNA 1347841 and 1348463 (RNASeq), PRJNA 1348465 and 1348470 (Microbiota).".

**Funding:** Funding for this project is from the Orthopedics Departmental Fund RUMC, no grant number (MMM), the Katz/Rubschlager Endowed Chair at RUMC (AP), and Versus Arthritis, UK Grant 22277 (AJD and CMM). None of the sponsors or funders had any role in the study design, data collection and analysis, decision to publish, or preparation of the manuscript.

**Competing interests:** The authors have declared that no competing interests exist.

a suppression of intestinal inflammatory and immune responses following the induction of joint inflammation and only minimal and transient changes in the microbiome. The results help clarify the molecular responses of intestinal tissues to inflammatory stresses that accompany the pathogenesis of inflammatory joint diseases.

## Introduction

Over the past decade, numerous studies have investigated the role of gut health in the progression of chronic musculoskeletal disorders, including rheumatoid arthritis (RA), spondylarthrosis, ankylosing spondylitis, and psoriatic arthritis [1,2], as well as osteoarthritis [3,4] and bone-related conditions [5], such as osteoporosis [6] and implant loosening [7]. Many of these studies have identified alterations in the composition of the gut microbiota—collectively referred to as *dysbiosis*—characterized by reduced microbial diversity and/or the expansion of specific bacterial taxa that influence host physiology [8,9]. For instance, one study reported that 18% of RA patients experienced gastrointestinal discomfort and postprandial fullness compared to non-RA individuals, with nausea (OR 4.0, 95% CI 1.1–14.2) and stool leakage (OR 8.2, 95% CI 1.03–66) representing more severe symptoms [1]. Similarly, inflammatory bowel disease has been reported in approximately 5–10% of patients with ankylosing spondylitis [2].

However, since acute and chronic inflammatory disease states are accompanied by autocrine and paracrine production of mediators (i.e., cytokines, chemokines, and growth factors), activation of immune cells, and modification of enteric neuronal signals, they can transform the host intestinal environment by altering the structure and the intestinal barrier and function of its resident cells [3,4]. The intestinal barrier is composed of an outer mucus layer in contact with the commensal gut microbiota [5], anti-microbial proteins, secreted immunoglobulin A molecules [6], a specialized epithelial cell layer [7], and an inner lamina propria that is populated with innate and adaptive immune cells [8]. The multicellular intestinal epithelium consists of multiple cell types, including absorptive enterocytes, mucin-secreting goblet cells, enteroendocrine cells, Paneth cells, and intraepithelial lymphocytes [9]. Furthermore, specific interactions of structural molecules secreted by the epithelial cells, such as mucins and tight junction proteins provide selective permeability for nutrient absorption, while also providing a blockade as primary defense against bacteria to maintain immune homeostasis [10].

Disruption of intestinal epithelial barrier function has been reported as integral to rheumatoid arthritis (RA) progression and pathogenesis [11,12]. In fact, an early disease response in a collagen-induced model for RA (CIA) was shown to be through the Zonulin-CXCR3 mediated dissociation of tight junctions [13]. In addition, elevated HIF2α in intestinal epithelial cells during RA progression can accelerate disease progression. Therapeutic mitigation of those pathogenic responses protected intestinal barrier function and diminished activation of intestinal and lymphatic T-helper 1 (Th1) and Th17 [14]. However, it remains to be determined if the intestinal responses in the

RA animal model are mediated by RA- specific systemic autoimmune responses, and/or if they develop in response to bone and joint tissue destruction mediated by an inflamed synovium ("RA pannus").

In addition to RA [15,16], multiple reports suggest the existence of a "joint-gut communication axis" in osteoarthritis, spondylarthrosis, and degenerative disc disease [17–20] that can be part of disease initiation and/or progression. The gut microbiome is proposed as both a target and a mediator in this crosstalk. Likewise an increasing number of studies have explored a correlation between perturbed bone metabolism in osteoporosis, diabetes [21], fracture healing [22], and implant loosening [23] on intestinal homeostasis. These studies also focus predominantly on the role of intestinal dysbiosis and perturbed production of beneficial bacterial metabolites such as short chain fatty acids [15,24–26], with a few studies implicating the role of T-cells in the communication routes [27,28].

To further delineate the molecular and cell biological pathways in intestinal responses to joint inflammation, we used a rat model in which local joint inflammation is induced by intra-articular injection (IAI) of Complete Freund's adjuvant (CFA) [29,30]. The predominant inflammatory response of a single injection of the adjuvant into a joint is a rapid development of synovial hyperplasia and joint swelling with immune cell infiltration (macrophages, neutrophils) [31]. The model is widely used to study pain behavior and intra-articular cytokine production, and can produce changes in the intestinal microbiome [32,33]. It should be noted that although joint inflammatory parameters are similar to RA, the model lacks RA characteristics, such as development of systemic T-cell driven autoimmunity or polyarthritis [34–36].

We used this model in the context of our objective to determine whether inflammatory changes in the CFA-exposed synovium induced temporally related changes in the intestinal barrier of the ileum and/or the colon. The timeline of joint inflammation was assessed by measuring knee joint swelling, joint histology, synovial fluid levels of bikunin (a component of both the acute phase protein pre-alpha-inhibitor (PαI) and inter-α-inhibitor (IαI) [37], bulk-RNA sequencing of synovial tissue, and micro-computed tomography for trabecular and cortical bone changes [38]. At the same time points, intestinal tissue responses were assessed by examining changes in components of the intestinal barrier. This included histochemical analyses of mucus layers, alterations in the intestinal tissue transcriptome using bulk-RNA Sequencing, composition of the mucus layers and abundance and distribution of CD4 and CD8 positive T-cells. 16S RNA analyses for fecal microbiome composition in ileum and colon were also completed.

Taken together, our data show that the intestinal tissue response to an acute joint inflammation was region specific. The ileum responded with increased mucus secretion and redistribution of CD8 + intraepithelial lymphocytes (IEL) as well as a broad suppression of immune function and metabolism-related transcripts. In addition, concurrent silencing of similar pathways was seen in the colon. The intestinal cell response along with the minimal change in microbiome composition in either intestinal compartment indicated that as a result of an acute inflammation in the joint, intestinal cells will respond with allostasis [39]. This is a mechanism to defend against stress to protect barrier and immune functions as well as microbiota composition and thereby maintain overall function of the gut as a vital organ. This is discussed in the context of examining the mechanisms leading to failure of such preventative responses during progression of chronic inflammatory diseases, such as rheumatoid arthritis, to develop targeted interventions for mitigation of intestinal dysfunction in these diseases.

## Results

### Knee joint responses following IAI of CFA

To establish a time course of knee joint responses following CFA or PBS injections, CFA-IAI and naïve rats were sacrificed at days 3, 7 and 14 post-IAI for analyses (S1 Fig and Methods). Knee joint swelling in the medial-lateral orientation had developed by 3d post-CFA-IAI and remained elevated through 14d ($p < 0.001$, 1-way ANOVA) compared to Naïve and Saline controls (**Fig 1A**). There was also a significant time effect with CFA-IAI ($p < 0.001$, 1 way ANOVA).

A transient increase in serum protein effusion into the joint space was confirmed by significant increased synovial fluid contents between groups ($p < 0.001$, 1-way ANOVA) of serum-derived bikunin•CS as well as in both inter-α-inhibitor (IαI)

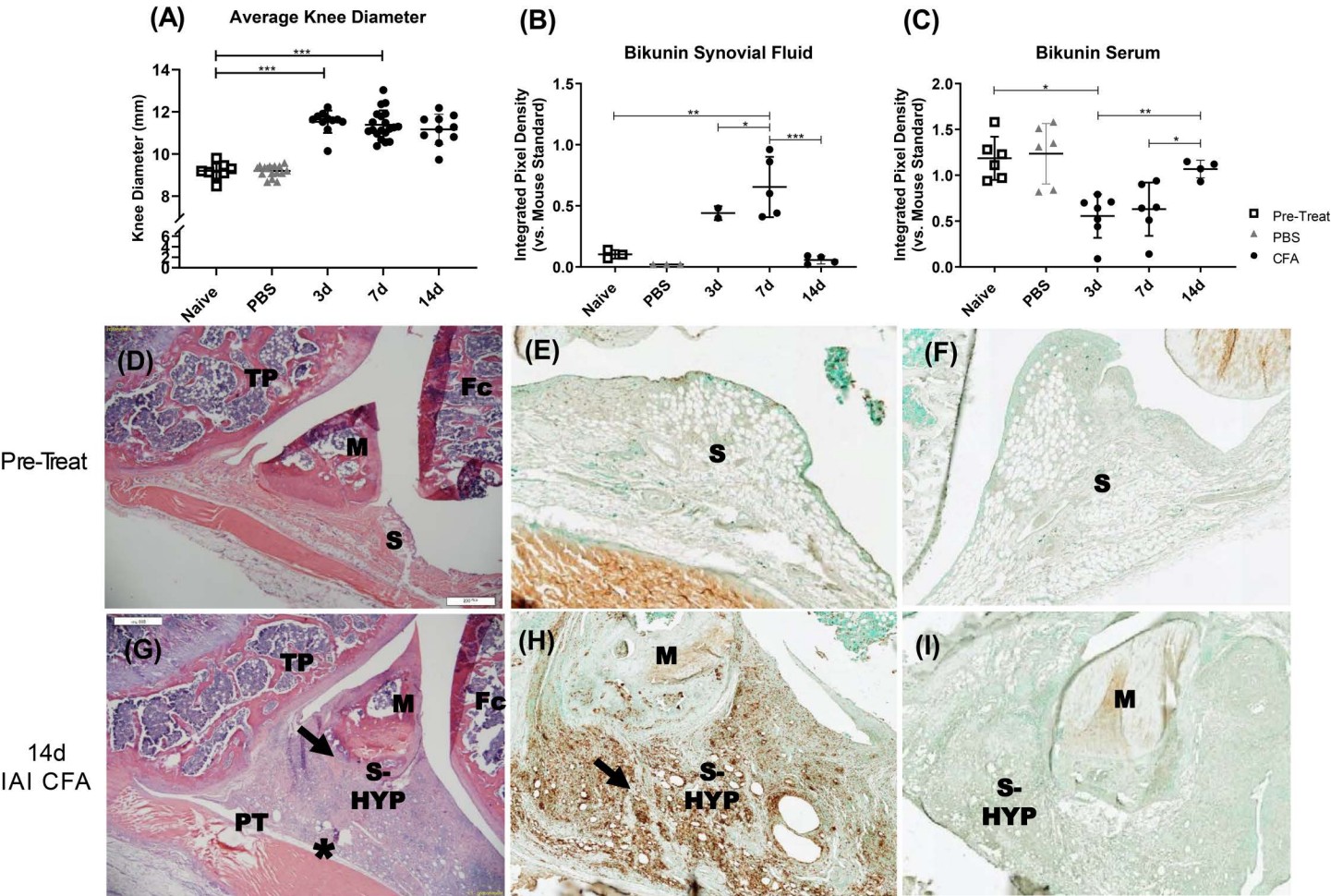

**Fig 1. Analyses of knee joint responses to IAI of CFA or PBS. (A)** At time of sacrifice, the diameters of knee joints were measured as described in the Method from rats without treatment (Pre-Treat), after IAI of PBS and 3 days (3d), 7 days (7d) and 14 days (14d) post-CFA. CFA-IAI knees showed significant swelling at all three time points, when compared to pre-treatment or PBS-IAI (***=p<0.001). **(B,C)** Abundance of bikunin species in synovial fluids (**B**) and sera (**C**) were determined by western blotting as described in the Methods and Supporting Information S2 Fig. Note that for pre-treatment (Pre-Treat) and PBS groups, the measurements at 0d and 14d, respectively, were combined since no significant differences were observed between those two groups (general linear model 0d v. 3d v. 14d PBS serum p=0.226). For A-C *=p<0.05, **=p<0.01, ***=p<0.001. Coronal formalin fixed paraffin embedded (FFPE) sections of pretreatment and 14d IAI-CFA knee joints were stained with H&E (**D & G**), anti-CD68 (**E & H**) or anti-CD4 (**F & I**). The CFA-induced synovial hyperplasia enriched in collagenous ECM and infiltrated by CD68+macrophages is indicated by black arrows and the swollen patellar tendon by (*). M=meniscus, Fc=femoral condyles, TP=tibial plateau, S=synovium, PT=patellar tendon.

and pre-α-inhibitor (PαI) (Fig 1B & S2B Fig, S2C). Notably, a concurrent significant decrease in all these species in serum (**Fig 1C**) at times of peak inflammation (p=0.005, 1-way ANOVA) suggested that the liver did not respond by upregulating production of these acute phase proteins in this model. Furthermore, a concurrent abundance of albumin and IgG in synovial fluids at 3d post-CFA-IAI (S2D Fig), as determined by SDS-PAGE confirmed the accumulation of serum derived components in the joint space. However, since these proteins were no longer present in the synovial fluids collected at 14d post-CFA, but increased knee diameters persisted, the latter was likely due to joint tissue responses, such as hyperplasia and fibrotic remodeling of the synovium and joint capsule, as well as patellar tendon swelling (Fig 1D-I). No alteration in joint swelling, joint fluid composition or tissue remodeling were noted after PBS-IAI.

## Transient changes in metaphyseal bone following CFA-IAI

As reported by others using the CFA rodent model [29,40], micro-CT analyses of the distal femoral metaphyseal and epiphyseal bone detected changes in both the cortical and trabecular bone compartments. Surface pitting was observed at 3d post-CFA-IAI in the proximal region of the patellar groove and by 7d the pitting extended to the periosteal surface of the epiphysis and distal metaphysis. (Fig 2A). Adjacent to the periosteal surface, periosteal reactions were present in n = 7 7d and n = 2 14d rats, primarily in the antero-mediolateral region of the distal femur (representative image, Figs 2B & S3 Fig). These reactions occurred transiently coincident with the peak knee swelling, metaphyseal bone loss, and synovial fluid content of bikunin in response to CFA-IAI. By 14d the cortical surface pitting had resolved, leading to the smooth appearance of the periosteal and patellar groove surfaces (Fig 2B–D). These cortical changes were associated with significant alterations in cortical bone geometry. Specifically, cortical thickness exhibited significant differences with an overall one-way ANOVA result (p = 0.003), and Bonferroni post-hoc analysis revealed significant changes between 3d and 7d (p = 0.022) as well as between 3d and 14d (p = 0.003). Medullary area also showed an overall significant effect (p = 0.031), with Bonferroni post-hoc significance observed between 3d and 14d (p = 0.041). Similarly, total area demonstrated significant overall changes (p = 0.036), with Bonferroni post-hoc significance between 3d and 14d (p = 0.047). Cortical area

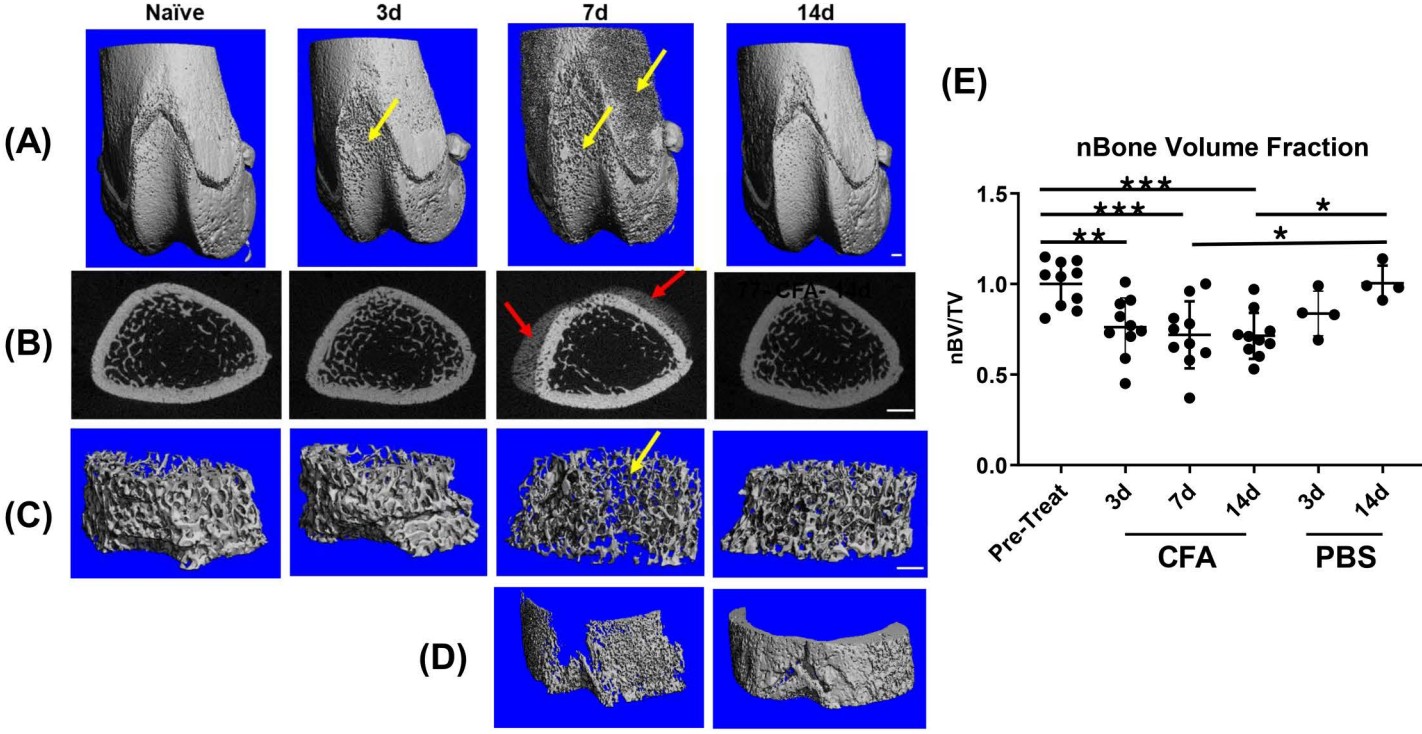

**Fig 2. µCT analyses of femoral metaphyses and epiphyses following IAI-CFA. (A)** 3D rendering of the distal femoral metaphysis and epiphyses show cortical pitting at 3d in the proximal region of the patellar groove with pitting extending to the epiphyseal and metaphyseal regions at 7d (yellow arrows). There is a cortical regenerative response visible by 14d. **(B)** Representative 2D slices through the distal metaphysis adjacent to the patellar groove show the typical appearance of a periosteal mineralization response (red arrows) in the antero-mediolateral region of the femur. This was present after CFA-IAI in (n = 7) 7d and (n = 2) 14d rats. **(C)** 3D renderings of the metaphyseal trabecular bone adjacent to the cortical bone showed decreased bone volume by 7d post-CFA, which also recovered by 14d. Scale bars all panels = 1 mm. **(D)** Representative 3D renderings of the isolated cortical periosteal mineralization response in the proximal femur at 7d and 14d post-CFA-IAI. The spatial distribution of the cortical pitting at 7d shown in panel A matches the periosteal reaction at 7d and is corrected by 14d post-CFA. **(E)** Bone volume fraction (BV/TV) normalized to pre-treatment group. BV/TV decreases with CFA-IAI compared to both pre-treatment and PBS-IAI. *=p<0.05, **=p<0.01, ***=p<0.001.

showed a trend toward significance (p = 0.069). Trabecular bone volume fraction, normalized to pre-treatment groups, was significantly decreased in the distal femoral metaphysis following CFA-IAI (Fig 2E). This reduction persisted through the 7d peak of inflammation and remained evident at 14d post-CFA-IAI.

## Transcriptomic responses of the knee joint synovium following IAI of CFA or PBS

Assessment of changes in tissue- or cell-specific transcriptomic landscapes by RNASeq methodologies has become an essential tool to identify and quantitate responses to disease-causing injurious stimuli in a range of tissues [41]. We have applied bulk-RNASeq methodology and post-sequencing bioinformatic analyses to identify such changes using the Database of Essential Genes (DEG) and associated pathway responses of synovial tissues to IAI of CFA or PBS vehicle control injection.

A global overview of gene expression profiles in synovium at pre- (0d) and post-CFA or PBS is shown in Fig 3A. Whereas PBS injections caused only minimal shifts (increased or decreased abundance) in gene expression profiles at early (3d) and or late (14d) time points, CFA-IAI resulted in extensive changes in the transcriptome, at all time points (3d, 7d, and 14d). DEGs at each post-IAI treatment time point, relative to 0d were calculated and significantly modified genes were

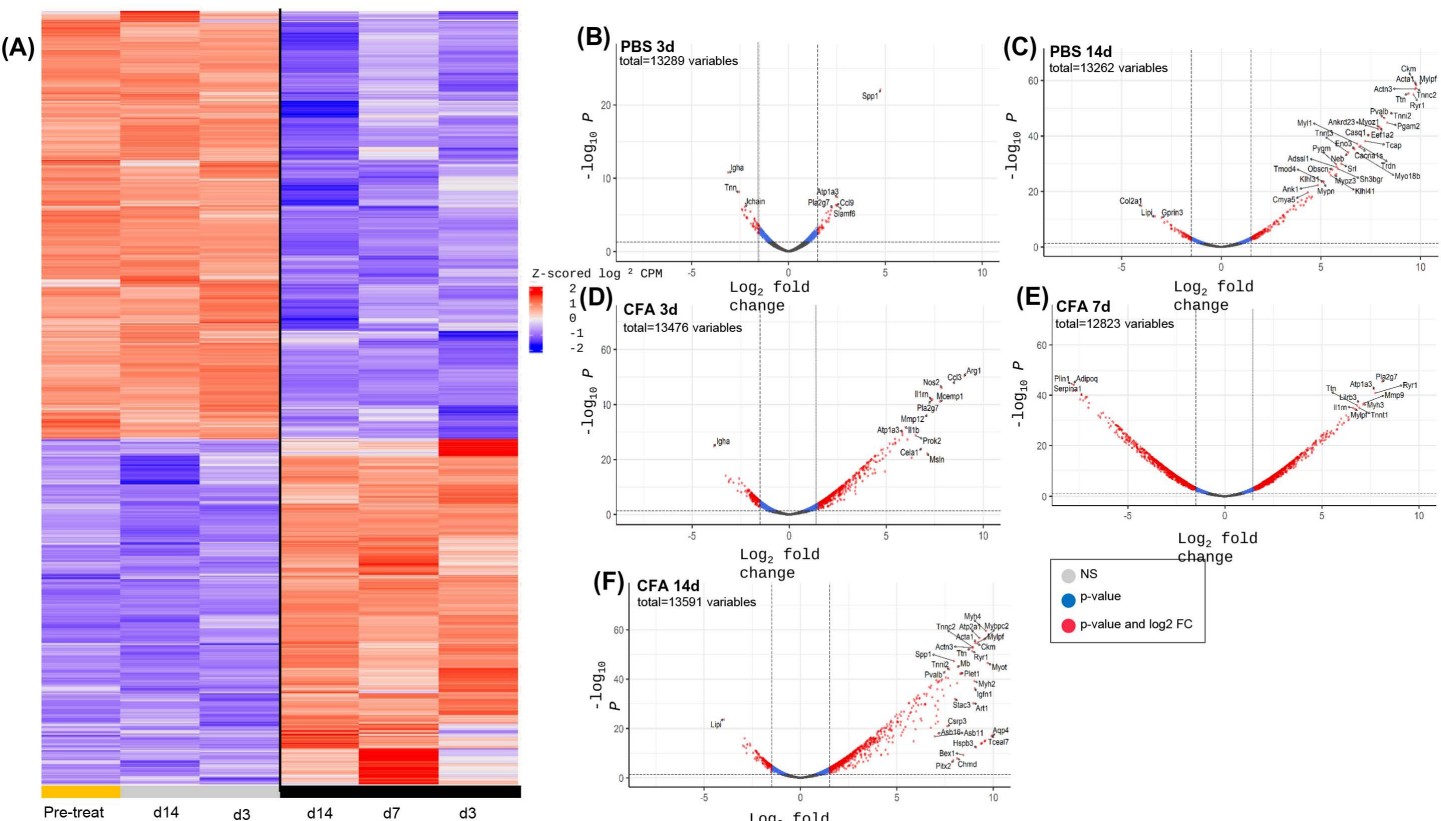

**Fig 3. Differentially expressed genes in synovial tissue at 3d, 7d and 14d after IAI of CFA or PBS.** (A) Heat map illustrating overall shifts in transcriptomic landscape after IAI-PBS (grey bars) or CFA (black bars), relative to d0 (yellow bar). (B-F) Statistical significance (p-value) vs. the magnitude of change (log FC) were calculated as described in the Methods and DEGs are summarized in volcano plots. Log FC thresholds of −2 and +2 and an adjusted p-value threshold of 0.05 are delineated by the dotted lines. Upregulated transcripts are on the right, downregulated transcripts are on the left of the lines, and statistically significant DEGs are above the horizontal lines. **Grey dots =** not significant, **Blue dots =** Genes p < 0.05, **Red dots =** Genes p < 0.05 and 2-fold change.

outlined in the volcano plots (Fig 3B-F; red dots = activated >1.5 $\log_2$ FC, p-value<0.05 or suppressed < −1.5 $\log_2$FC, p<0.05). Using 'significant fold change' selection parameters, CFA exposure upregulated transcript abundance for 592, 1037, and 998 genes at 3, 7 and 14d, respectively, but decreased transcripts for 167, 998 and 166 genes at the corresponding time points. By comparison, after sham-IAI-PBS, increases relative to 0d occurred for 32 and 180 genes and suppression for 37 and 67 genes 3d and 14d, respectively.

The acute (3d) response to CFA-IAI resulted in limited but strong immune and inflammation responses, with >100 fold increases in transcripts of the neutrophil protein *S1009* [42], the macrophage receptor *Clec5a* [43] and the pro-angiogenic protease *Mmp9* [44] as well as the lymphocyte activating factor *Slamf6* [45]. In addition, multiple genes, previously identified with inflammatory arthritis showed sustained modification up to 14d. These included transcripts for pro-inflammatory mediators *Ccl3, Cxcl1, Ccl2, Ccl9, Nos2, Arg1, Il1m, Il1b,* and *Osm,* their regulators *S100a9* [46], *Slamf6* and *Il1r2*, the C-type lectins *Clec5a* and *Clec4e* [47], as well as *Lcn2* associated with bone damage in RA [48,49]. In addition, macrophage activating factor *Gdf15* [50], *Adam8* and an integrin, *Itgb8*, from synovial fibroblasts [51] were affected. Notably, transcripts of the *Spp1* gene that encodes the bone protein OPN (osteopontin) and is associated with RA-related synovial inflammation [52], were highly activated in the acute joint response at 3d to both, CFA- and PBS-IAI and remained high up to 7d post-CFA-IAI. Furthermore, transcripts for multiple genes with known functions in innervation and inflammatory pain, *Kcne5, Grin3a, Map2, Erc2, Serpini1, Gprin3, Trhde,* and *Mdga2* were suppressed at all investigated times post-CFA-IAI.

## Transcriptomic responses of the proximal and distal ileum and distal colon following IAI of CFA

Bulk RNASeq analyses were also performed for intestinal tissues collected at 3, 7 and 14d post-CFA-IAI from the proximal and distal ileum and the distal colon. Bioinformatic analyses of sequencing data for DEGs, pathways, and biological process modifications were performed as described in the methods. It should be noted that since PBS-IAI did not yield significant modification of synovial gene transcriptions when compared to CFA exposure (Fig 3), RNASeq analyses of gut tissues were only performed on naïve and CFA-treated gut samples. Gene expression profiles and DEGs for all three intestinal regions (proximal and distal ileum and colon) are shown in Figs 4−6, respectively. These show a wide range of transcriptional changes relative to d0, with statistically significant increases and decreases in DEGs (<$\log_2$−1.5 and> $\log_2$+1.5). To supplement the volcano plots in B-D panels, details of DEGs are also shown as heat maps (S3A Fig-S3C Fig & S5A-S5C Fig). Together, these data show that changes in affected genes were region-specific and varied with time post-CFA treatment. For example, maximum modifications of the proximal ileal transcriptome were seen at 7d post-CFA, compared to 3d for the colonic transcriptome, with the distal ileum showing a weaker and rather uniform response at all 3 post-CFA time points (S3 Fig).

Transcripts that were modified at more than one post-CFA time point in each of the three intestinal regions are summarized in S4A-S4C Fig, and many of these DEGs implicate responses of several cell types in the intestinal barrier. For example, in the proximal ileum, the *Mptx1* pseudo gene [53,54] is a gut-specific member of the pentraxin family, which is involved in bridging adaptive and innate immunities in inflammatory responses in enterocytes. It is a product of Paneth and epithelial cell transcripts for several REG family proteins, *Reg1a, Reg3b and Reg3G* [55,56], which are multifunctional molecules secreted by Paneth cells to exert anti-apoptotic, anti-inflammatory and anti-microbial effects were decreased after CFA-IAI. Two genes, *Adgrg2* [57] and sodium and potassium absorption, *Tmprss15* [58], are known to be part of Tuft cell function in the regulation of immune responses. In addition, transcripts for two genes associated with neutrophil responses, *Mpo* [59] *and Mst1* [60], and known to be involved in IBD pathologies, were elevated at 7d and 14d post-CFA-IAI.

In the distal ileum, in addition to the increased transcript for the *Mptx1*, a strong and sustained response of enterocytes was supported by increases in the transcription factor *Hoxa9* [61], the amino acid transporter *Slc7a14* [62], the tight junction protein *Cldn8* [63], the anti-microbial Lysozyme *Lyc2* (*Lyzl-1* [64]), and the *Rdh7* gene, for retinoic acid metabolism [65]. Downregulated expression was seen for epithelial cell specific genes, involved in glucose transport and uptake, *Gpr17* [66], *Slc2a2* [67] and *Slc5a4* (*Sglt3*) [68].

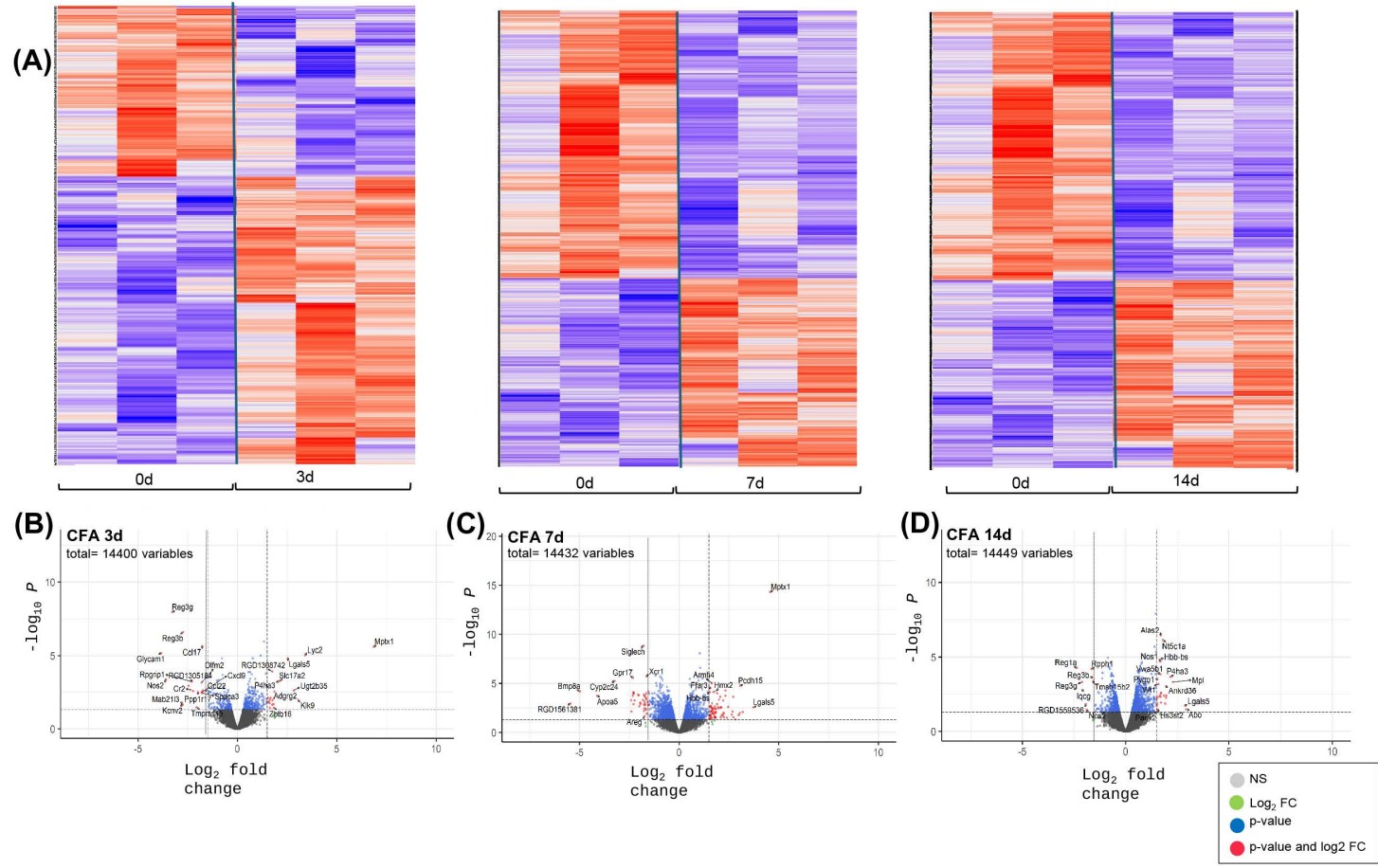

**Fig 4. Differentially expressed genes in proximal ileum tissue at 3d, 7d and 14d after IAI of CFA. (A)** Heat maps illustrating overall shifts in transcriptomic landscape after IAI of CFA relative to d0. Each vertical column corresponds to tissue from a single animal. **(B-D** CFA at 3d, 7 d and 14d respectively) Statistical significance (p value) vs the magnitude of change (log FC) are summarized in volcano plots using the log FC thresholds of −2 and +2 and P-values of 0.05 (dotted vertical and horizontal lines). **Grey dots =** not significant, **Green dots =** Genes 2-fold change, **Blue dots =** Genes p < 0.05, **Red dots =** Genes p < 0.05 and 2-fold change.

The most profound transcriptional response to CFA-IAI was seen in the distal colon with decreased mRNA levels for a wide range of genes (S3C Fig and S4C Fig). Downregulated transcripts were seen for genes involved in regulation of intestinal immune responses, including the immune cell infiltration-related *Serpine1* [69] and the transcription factor *Atf3* [70,71]. Several of the modified transcripts also point to cell-specific responses in the colon including the epithelial cell-associated genes *Btnl8* [72], *Fabp1* [73], *Prkg2* [74], *Car1* [75] and *Dmbt1* [76]; the fibroblast/myofibroblast associated genes *Cxcl11* [77], *Nr4a1* [78], and *Tbx5* [79]; as well as the enteroendocrine cell gene *GcG* [80]. Small, but significant increases were also seen for the transcription factor *EyA1* [81], which is implicated in the activation of cell proliferation and EMT in colon cancers.

## Pathway and biological process allocation of DEGs using GO enrichment

To assign a functional significance of modified gene transcripts within the context of cellular pathways in synovium and intestinal tissues, GO enrichment analyses was performed using the KEGG pathway database. Individual pathways were further assigned to the following six biological processes listed in that database: Metabolism, Cellular Processes, Genetic

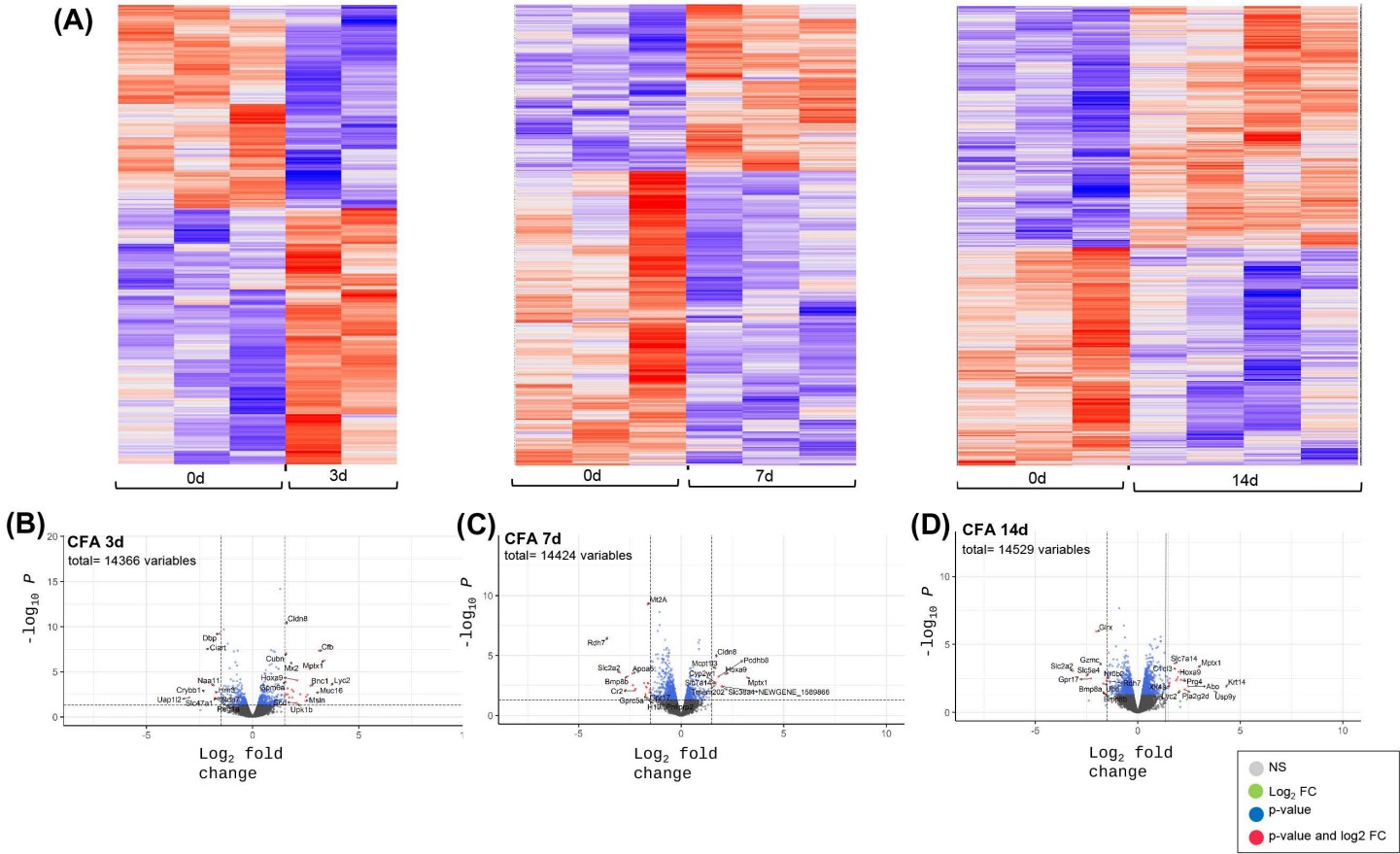

**Fig 5. Differentially expressed genes in distal ileum tissue at 3d, 7d and 14d after IAI of CFA. (A)** Heat maps illustrating overall shifts in transcriptomic landscape after IAI of CFA relative to d0. Each vertical column corresponds to tissue from a single animal. **(B-D** CFA at 3d, 7 d and 14d respectively)** Statistical significance (p value) vs the magnitude of change (log FC) are summarized in volcano plots using the log FC thresholds of −2 and +2 and P-values of 0.05 (dotted vertical and horizontal lines). **Grey dots =** not significant, **Green dots =** Genes 2-fold change, **Blue dots =** Genes p < 0.05, **Red dots =** Genes p < 0.05 and 2-fold change.

Information Processing, Environmental Processing, Immune System and Nervous System. Note that all gene sets with a log FC value of <−0.3 and>+0.3, and an adjusted p-value <0.05, were included. The data are displayed in dot plots for synovium (S5 Fig & S6 Fig), proximal ileum (S7 Fig), distal ileum (S8 Fig) and distal colon (S9 Fig) and are also summarized in Venn diagrams (Figs 7 & 8) Following CFA-IAI, a large number of pathways were activated in synovial tissue (Fig 7A & S5A Fig) (i.e., 45, 40 and 37 at 3d, 7d and 14d, respectively) or suppressed (i.e., 22, 32 and 6 at 3d, 7d and 14d, respectively). As expected from previous reports of the CFA model [29,31], the adjuvant produced robust and persistent immune and inflammatory reactions in the synovium. Activated pathways included "NK cell mediated cytotoxicity", "Neutrophil extracellular trap formation", "Th1 and Th2 cell differentiation", "Th17 cell differentiation" and "B cell receptor signaling" pathways at 3d, 7d, and 14d post-CFA-IAI. This was accompanied by a stimulation of "Toll-like-receptor"-, "C-type lectin-" "IL17-" and "NOD-like"-immune cell signaling characteristic of immune cells. Other inflammation-related pathways such as "cytokine–cytokine receptor interactions", "chemokine signaling" and "NF-kB signal transduction" were also stimulated. Throughout the post-CFA period examined, a range of other pathways were suppressed, such as the Hippo pathway [82,83], which can lead to induction of cell division and the suppression of apoptosis by PPAR signaling. This would be expected to interfere with control of the inflammatory response to TNF-α in synovial cells [84]. Other

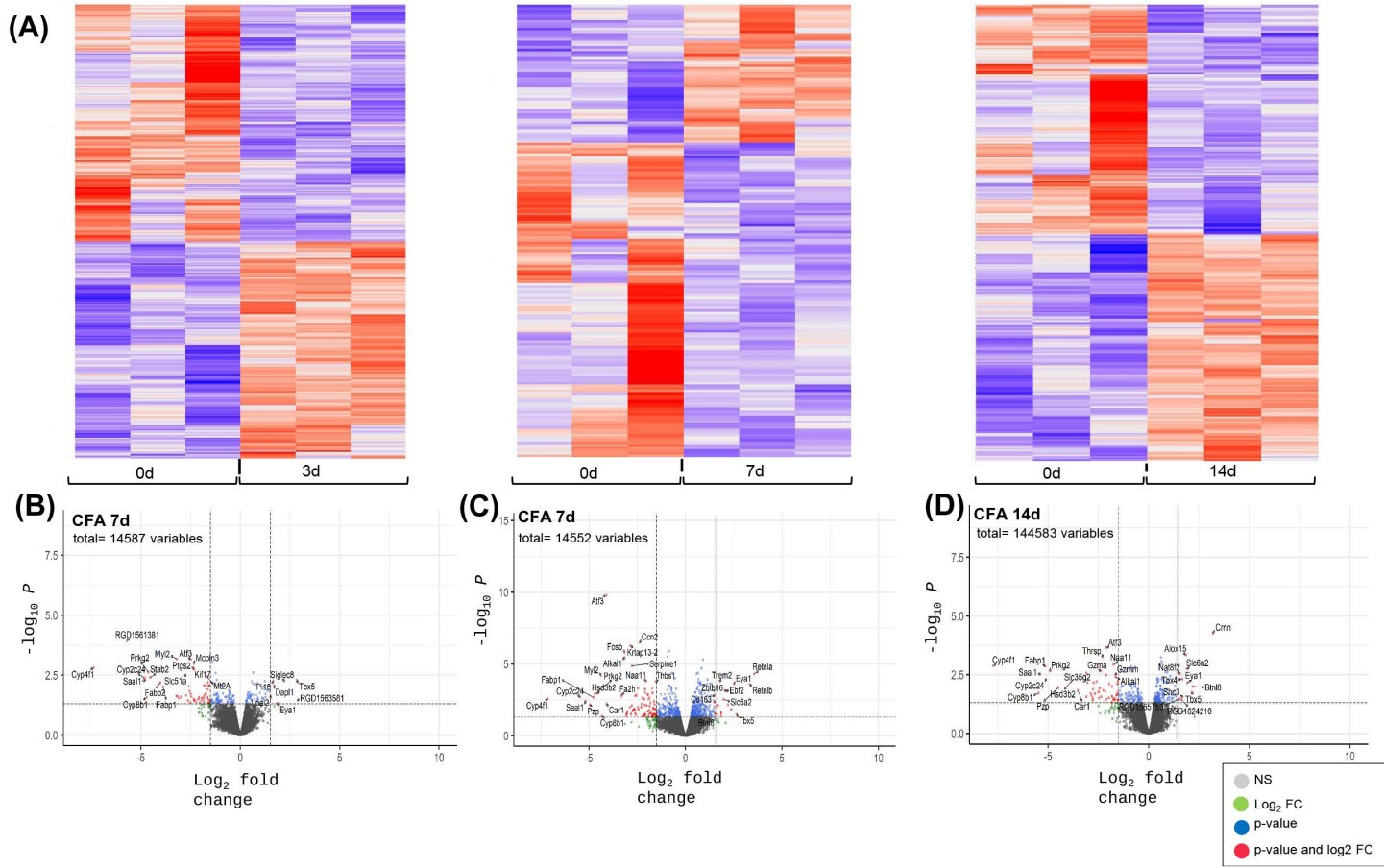

**Fig 6. Differentially expressed genes in distal colon tissue at 3d, 7d and 14d after IAI of CFA. (A)** Heat maps illustrating overall shifts in transcriptomic landscape after IAI of CFA relative to d0. Each vertical column corresponds to tissue from a single animal. **(B-D** CFA at 3d, 7 d and 14d respectively)** Statistical significance (p value) vs the magnitude of change (log FC) are summarized in volcano plots using the log FC thresholds of −2 and +2 and P-values of 0.05 (dotted vertical and horizontal lines). **Grey dots=**not significant, **Green dots=**Genes 2-fold change, **Blue dots=**Genes p<0.05, **Red dots=**Genes p<0.05 and 2-fold change.

notable adjuvant-activated responses were seen in lysosomal, phagosomal, apoptotic, and necroptotic pathways, while neuronal system pathways such as "axon guidance" and "glutamatergic synapse function" were suppressed. By comparison, PBS-IAI resulted in a very minimal response in the synovium. The only modified pathways post-PBS-IAI that showed overlap with those modified by CFA-IAI were "heparan sulfate biosynthesis", "RNA polymerase", and "tryptophan metabolism" at 3 d and "neuroactive ligand-receptor interaction" and "HIF-1 and calcium signaling" pathways at 14d. There was a lack of immune or inflammation-related pathways changed by PBS. This is in contrast to what was expected and previously shown as a mild sham-IAI response in the clinical setting for sham injection [85] or joint space lavage [86].

The corresponding GO analyses of intestinal tissues (Fig 8 & S6-S8 Fig) showed fewer affected pathways than those identified in the synovium. As expected from the extensively downregulated transcriptome (Fig 4–6) after CFA-IAI, many of the intestinal pathways were suppressed, most notably those associated with the immune system. In the proximal ileum, "Th1 and Th2 cell differentiation", "Natural killer cell mediated cytotoxicity", "Th17 cell differentiation", and "Antigen processing and presentation" were suppressed over the entire experimental timeline. In the distal ileum, the "Intestinal immune network for IgA production" and the "Antigen processing and presentation" pathways were suppressed at 7d

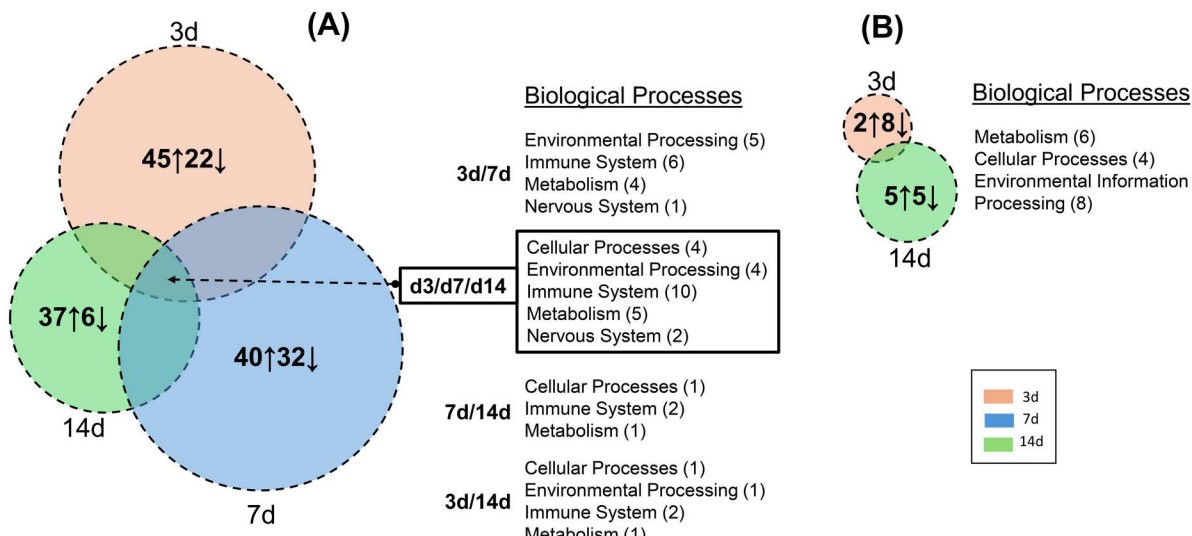

**Fig 7. Venn diagram illustration of modified biological process in synovium.** Following **(A)** CFA or **(B)** PBS exposure. The numbers in individual circles indicate the total pathways activated (↑) or suppressed (↓) at each time point, and these are individually displayed and in color-coded KEGG-based grouping of "Biological Processes". Processes modified at two (3d/7d, 7d/14d, 3d/14d) or three (3d/7d/14d) times points are listed to the right of the Venn diagram and the number of affected pathways in each Process is shown in brackets.

and 14d. These immune pathways were also suppressed in the distal colon, together with "B cell receptor" and "T cell receptor-signaling, and "Leukocyte trans endothelial migration". Transcripts for a range of metabolic pathways and signaling pathways were also decreased in both the ileum and colon.

Additional RNASeq data analyses to define modulation of Myeloid or Lymphoid-specific pathway using the GSEA database against the msigDB Gene Ontology gene sets (subset BP) are shown in Fig 9. The majority of identified GOs, i.e., 22, were associated with cells from the myeloid lineage, 11 of those were for leukocytes and 5 for dendritic cells. All pathways, except for "Myeloid Development" in the distal colon, were downregulated relative to naïve intestinal regions. The number of affected pathways with decreased transcript abundance was largely in the proximal ileum, fewer in the distal colon, but largely unaffected in the distal ileum. Moreover, pathway modulations were most pronounced at 3d in all regions but largely restored to pre-injury levels by 14d post joint injury. Lymphoid pathways were minimally affected. "Lymphoid organ development" was suppressed in the proximal ileum and lymphoid progenitor cell differentiation activated at d14 in the distal colon

A comparison of pathway-specific responses between synovial and intestinal tissues, is shown in **Fig 10**. Although many of the same pathways were affected, activation responses in the synovium gave corresponding suppression responses in the intestinal tissues, and this was particularly evident during the peak joint inflammation responses at 3d and 7d. This, together with the DEG data, confirms a generalized 'silencing' of inflammatory and immune reactions in the intestine post-CFA-IAI. This likely constitutes a shielding response to maintain intestinal barrier function in the presence of an acute innate immune and inflammatory insult at a remote tissue site.

### Effect of IAI CFA on intestinal immune cells and mucin metabolism

To test for alterations in intestinal immune cell population, such as macrophages and T-cells, we performed IHC analyses of sections from Swiss-rolled ileum and colon using anti-CD68 for macrophages, anti-CD4 and anti-CD8 for resident T cells (Fig 11A-C and S10-S12). In healthy tissue, CD68+ macrophages were almost exclusively located in the lamina propria of the ileum and the colon, and there was no change in their abundance or localization along the ileum. However, in

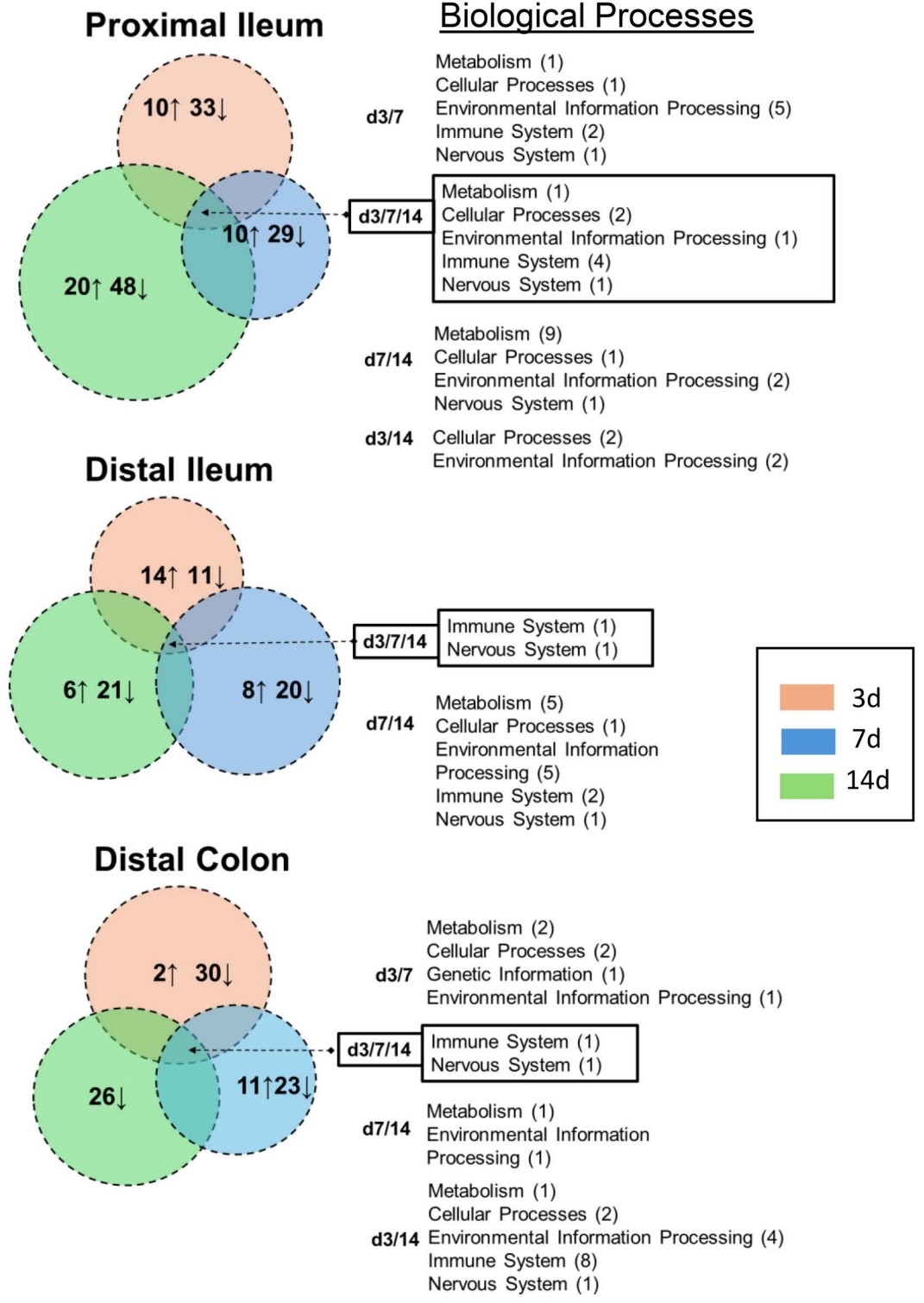

**Fig 8. Venn diagram illustration of modified biological processes in proximal and distal ileum and colon following CFA-IAI.** Numbers in individual circles indicate the total pathways activated (↑) or suppressed (↓) at each time point, and these are individually displayed and in color-coded KEGG-based grouping of "Biological Processes". Processes modified at two (3d/7d, 7d/14d, 3d/14d) or three (3d/7d/14d) times points are listed to the right of the Venn diagram and the number of affected pathways in each Process is shown in brackets.

# Biological Processes Gut Regions GO IDs

### MYELOID

| Biological Processes | PI | DI | CO | GO IDs |
|---|---|---|---|---|
| Common Myeloid progenitor cell proliferation | ↔ | ↓ | ↓ | GO:0035726 |
| Myeloid cell activation involved in immune response | ↓ | ↔ | ↓ | GO:0002275 |
| Myeloid cell differentiation | ↓ | ↔ | ↔ | GO:0030099 |
| Regulation of Myeloid cell differentiation | ↓ | ↔ | ↔ | GO:0045637 |
| Positive regulation of Myeloid cell differentiation | ↓ | ↔ | ↓ | GO:0045639 |
| Myeloid cell development | ↔ | ↔ | ↑ | GO:0061515 |

**Leukocyte**

| Biological Processes | PI | DI | CO | GO IDs |
|---|---|---|---|---|
| Regulation of Myeloid Leukocyte differentiation | ↓ | ↔ | ↔ | GO:0002761 |
| Positive regulation of myeloid leukocyte differentiation | ↓ | ↔ | ↓ | GO:0002763 |
| Myeloid Leukocyte differentiation | ↓ | ↔ | ↓ | GO:0002573 |
| Myeloid Leukocyte activation | ↓ | ↔ | ↓ | GO:0002274 |
| Regulation of myeloid leukocyte mediated immunity | ↓ | ↔ | ↔ | GO:0002886 |
| Myeloid Leukocyte migration | ↓ | ↓ | ↓ | GO:0097529 |
| Myeloid Leukocyte mediated immunity | ↓ | ↓ | ↓ | GO:0002444 |
| Positive regulation of myeloid leukocyte mediated immunity | ↓ | ↔ | ↔ | GO:0002888 |
| Myeloid Leukocyte cytokine production | ↓ | ↔ | ↓ | GO:0061082 |
| Positive regulation of myeloid leukocyte cytokine production involved in immune response | ↓ | ↔ | ↓ | GO:0061081 |
| Negative regulation of myeloid leukocyte mediated immunity | ↓ | ↔ | ↔ | GO:0002887 |

**Dendritic Cell**

| Biological Processes | PI | DI | CO | GO IDs |
|---|---|---|---|---|
| Myeloid Dendritic cell activation | ↓ | ↔ | ↔ | GO:0001773 |
| Regulation of myeloid Dendritic cell activation | ↓ | ↓ | ↓ | GO:0030885 |
| Negative regulation of myeloid Dendritic cell activation | ↓ | ↔ | ↓ | GO:0030886 |
| Myeloid Dendritic cell differentiation | ↓ | ↔ | ↔ | GO:0043011 |
| Myeloid Dendritic cell activation involved in immune response | ↓ | ↔ | ↓ | GO:0002277 |

### LYMPHOID

| Biological Processes | PI | DI | CO | GO IDs |
|---|---|---|---|---|
| Hematopoietic or Lymphoid organ development | ↓ | ↔ | ↔ | GO:0048534 |
| Lymphoid progenitor cell differentiation | ↔ | ↔ | ↑ | GO:0002320 |

**Fig 9. Myeloid and Lymphoid-associated pathway analyses using GSEA using against the msigDB Gene Ontology gene sets, subset BP.**
Activated pathways are marked with a red upward pointing arrow (↑) and suppressed pathways with a blue downward pointing arrow (↓). No change is indicated by a horizontal double arrow (↔). PI = proximal ileum, DI = distal ileum, CO = Colon.

the colon, at 3d post-CFA-IAI, transient increases in CD68 + ve cells surrounding crypts and interspersed between enterocytes along the villi was seen (black arrows, Fig 10A) was seen. This acute buildup of macrophages in the colon was further supported by increased transcript abundance of macrophage-specific genes *Trem2* [87] and *C1qc* [88] (Fig 10D).

 CD4 + T cells were distributed along the villi in the ileum and additionally in the apical regions in the colon, likely expressed by the intraepithelial lymphocytes [9] (IELs), which were unchanged during the post-CFA period (Fig 11B). In naïve intestinal tissue, the anti-CD8 also stained predominantly IELs (Fig 11C). However, at 3d and 7d post-CFA-IAI, increased staining of such cells in the apical part of the villi was seen in many regions throughout the ileum and the colon. In addition, at 7d CD8 + ve populations had also accumulated in the crypts (black arrows, Fig 11C). By 14d, however, the abundance of these T-cells and their locations had returned to those seen in naïve intestinal tissues.

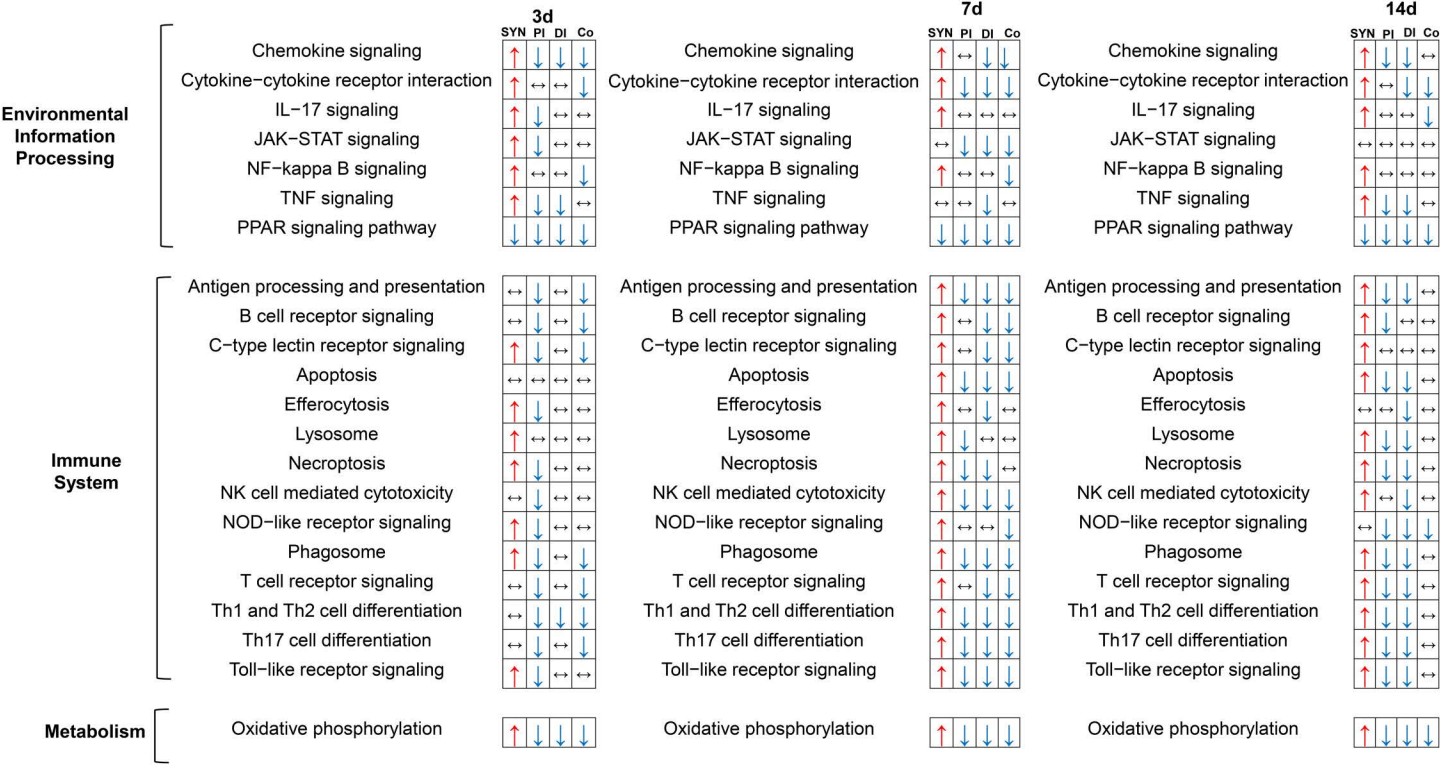

**Fig 10. Comparisons of modulated pathways in synovial and intestinal tissue following CFA-IAI.** Activated pathways are marked with a red upward pointing arrow (↑) and suppressed pathways with a blue downward pointing arrow (↓). No change is indicated by a horizontal double arrow (↔). SYN = synovium, PI = proximal ileum, DI = distal ileum, Co = Colon.

In keeping with the protective response of intestinal tissues to knee joint inflammation, there were no changes in cell proliferation (Ki67 staining) in any intestinal region (S13 Fig). We also observed increased secretion of mucins in the distal ileum, during the acute response phase at 3d & 7d, shown by Alcian blue staining (Fig 12A), and supported by increases in transcript abundance for the gel-forming mucins *Muc2* and *Muc5b* [89] (Fig 12C). Furthermore, histochemical staining with O-linked oligosaccharide specific lectins (GSL I, MAL II, and UEA) showed an increased reactivity in mucus with the GSL I (specific for terminal α-N-acetylgalactosamine or α-N-galactose residues) [90] in the distal ileum at 3d post-CFA-IAI (Fig 11B). There were no detectable changes in staining with MAL II (terminal α-2,3 sialic acid) or UEA (α-linked fucose) (S14 Fig). Modulation of mucin glycosylation in response to the joint inflammation was further underlined by altered mRNA transcripts for multiple glycosyltransferases, including *St6galnac1* (Fig 11C), the major sialyltransferase in goblet cells, known to be induced by microbial pathogens and critically important for maintaining mucus integrity [91].

### Effect of CFA-IAI on digestive functions and microbiome composition

Based on our RNASeq data, several digestive system pathways throughout the different regions of the intestine were affected by the inflammatory response of the knee joint. For example, transcripts encoded by genes associated with 'mineral absorption' were modulated in all 3 regions of intestine and were largely decreased compared to pre-treatment. The distal colon showed suppression of the fat digestion pathways, with decreases in expression of 4 associated genes (Fig 12C).

Notably, there was no evidence for development of dysbiosis in the model. The fecal microbiota from the distal ileum and colon were minimally altered in response to the knee joint inflammation and respective intestinal cell responses.

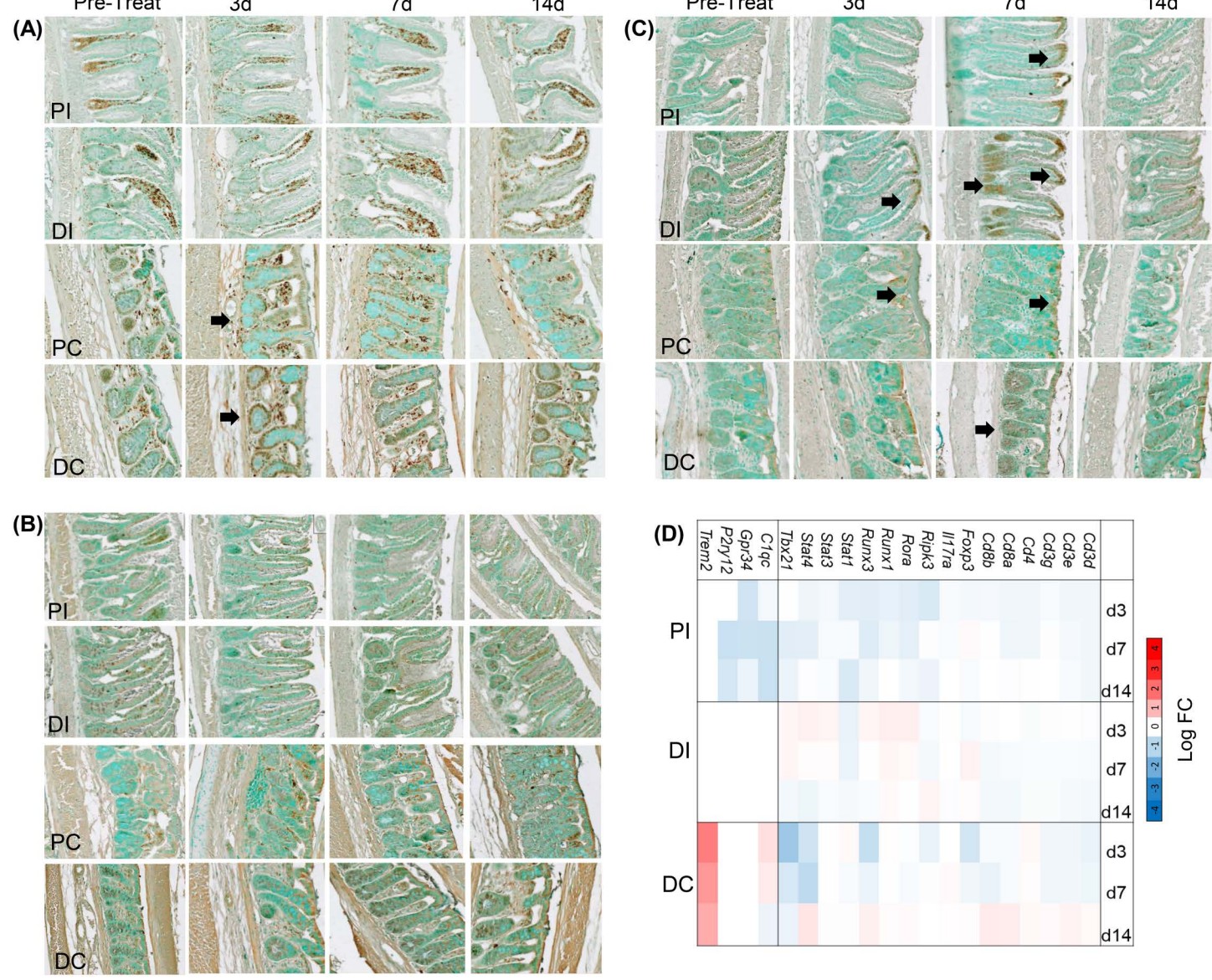

**Fig 11. Effect of CFA-IAI on macrophage and T cell responses in distal Ileum and colon.** Adjacent FFPE-sections from proximal and distal regions of the ileum and colon were stained with (**A**) anti-CD68 to localize macrophages or (**B**) anti-CD4 and (**C**) anti-CD8 to localize T-cells. Black arrows indicate regions of increased reactivity post CFA. (**D**) DEGs typically associated with intestinal macrophages or T cells caused by CFA-IAI are summarized in the heat map. PI = Proximal Ileum, DI = Distal Ileum, PC = Proximal Colon, DC = Distal Colon.

Alpha diversity for within-group comparison was unaltered, as shown by evenness (p = 0.697), Shannon index (p = 0.997), and Simpson index (p = 0.998). Relative bacterial abundance (>1%) for the distal ileum and the colon (Fig 12B) at the phylum level showed a slight shift from Naïve to 3d, 7d and 14d post-CFA-IAI. Amongst this shift, there was a decrease in *Romboustia* in the ileum at 3d post-CFA, followed by an increase at 7d and a return to naïve levels by 14d. *Lactobacilus* showed the opposite shift with an increase by 3d, decrease by 7d, and a return to naïve levels by 14d. This change in abundance supports a mild and importantly, a limited transient response of the microbiome when acute inflammation in

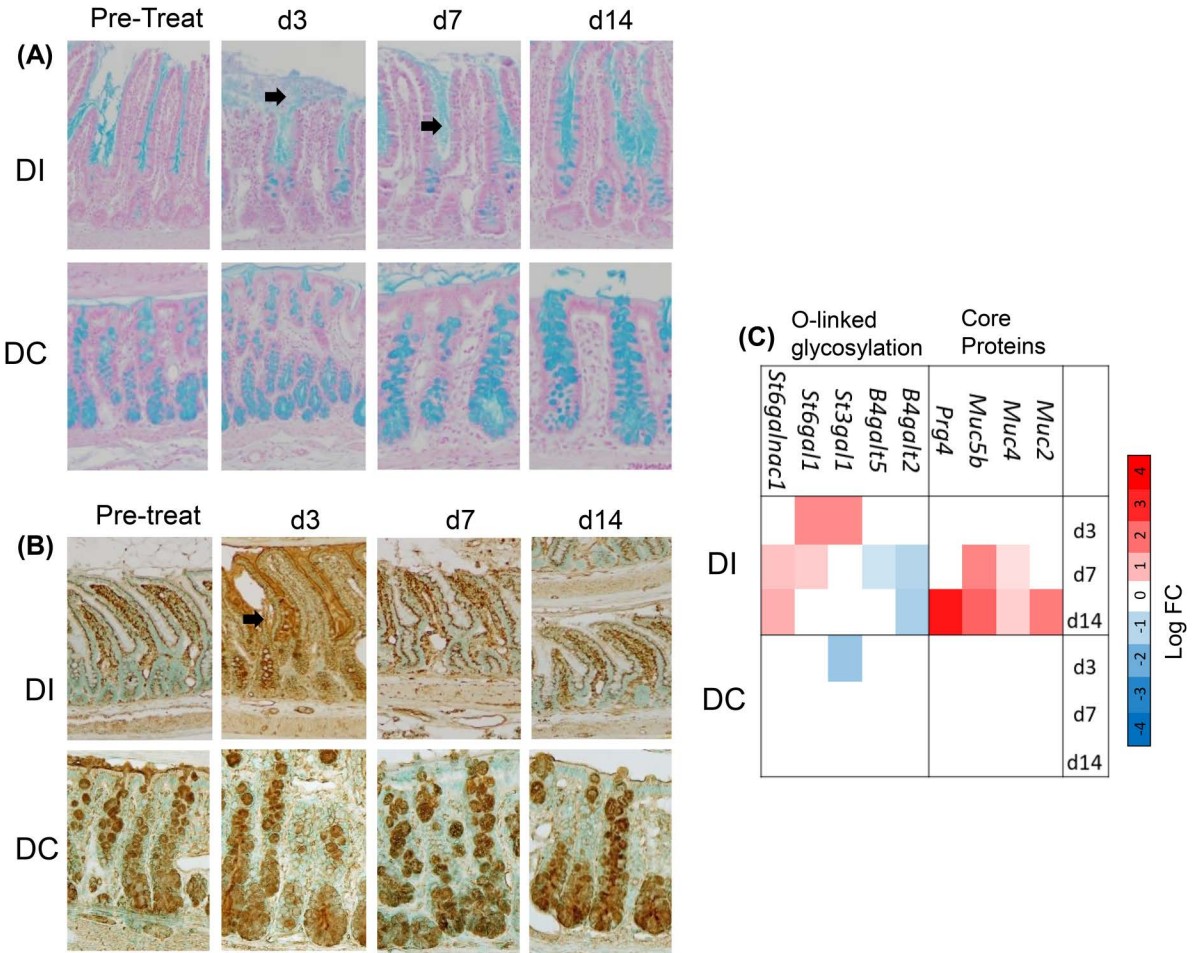

**Fig 12. Modified mucin secretion in distal ileum and colon following CFA-IAI.** Adjacent FFPE-sections of Swiss-rolled ileum or colon stained with Alcian Blue (**A**) or the biotinylated GSI lectin (**B**) as described in the Methods. Brown = positive stain; Black arrows show the increased mucin secretion with enhanced GSI staining in the DI at 3d post IAI of CFA. Modified gene transcripts in the mucin biosynthesis pathways in proximal and distal regions of the ileum and the distal colon, which were affected by IAI of CFA are summarized in the heat map in panel (**C**). PI = Proximal Ileum, DI = Distal Ileum, PC = Proximal Colon, DC = Distal Colon.

the joint peaks at 7d post-CFA-IAI. Due to the observation of a transient change in mucus layer 16S data were mined for the abundance of mucin-degrading bacteria (*Akkermansia, Alistipes*) and sialic acid cleaving bacteria (*Bifidobacterium*) (Fig 13 & S14 Fig). Apart from the appearance of low levels of *Akkermansia* in the feces from distal ileum and distal colon post IAI of CFA, the joint inflammation had no detectable effect on the other two species.

## Discussion

In this study, we present novel data on coordinated cell and molecular biological changes in the synovium and intestinal tissues in the acute CFA-induced knee joint inflammation model. Due to its reproducibility and rapid inflammation responses in the injected joint, this model has been used extensively for preclinical efficacy testing of a wide range of anti-inflammatory [92,93] and analgesic [94–97] RA drugs. A number of these studies have chosen the inflamed synovium as a target, with a focus on the cytokines and the NFκB signaling pathway [98–101]. This model has also been used in several studies to examine the effect of joint inflammation on the gut microbiota [32,94] supporting the existence of a

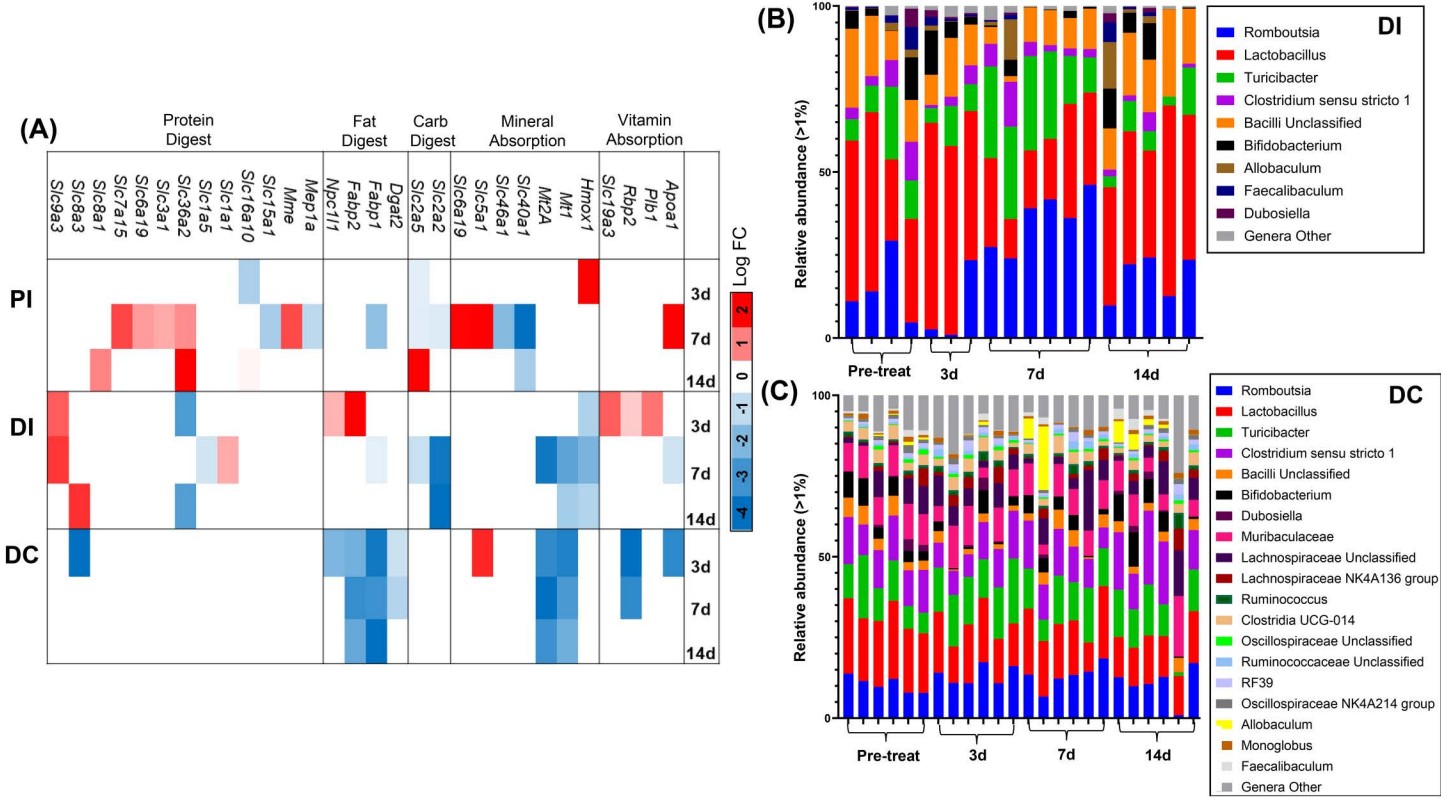

**Fig 13. Effects of CFA-IAI on genes in associated with digestive functions and microbiome compositions in distal ileum and colon. (A)** Data were obtained from the DEGs from analyses of RNASeq data (see Figs 3 & 5) and are shown as log fold changes (p < 0.05) as a heat map. **(B & C)** 16S Amplicon Sequencing of Fecal Microbiome was performed as described in the Methods. Relative bacterial abundance (>1%) is shown for the distal ileum (DI, **B**) and the distal colon (DC, **C**) at pre-treatment, 3d, 7d and 14d at the phylum level. Individual phyla are color-coded and listed on the right.

"cross-talk" between joint and remote tissues like the gut. However, only limited data of the cell biological responses in the intestinal barrier tissues and innate immune cell populations are currently available. Our data reported here provide a detailed and comprehensive assessment of cellular and molecular biological changes underlying inflammatory responses during the formation of a synovial pannus and corresponding intestinal responses.

Our data confirm an early and robust response of the joint to the adjuvant at 3d and 7d post-CFA-IAI. In addition to measurements of joint swelling, we showed joint effusion by elevated levels of serum albumin and serum-derived bikunin-containing complexes [37] in the synovial fluids. The latter include the liver-derived IαI and the acute phase proteoglycan PαI, which are known to accumulate in synovial fluids in osteoarthritis and rheumatoid arthritis [102–104]. In this context, it is notable that in an inflammatory tissue environment, bikunin-containing complexes can act as donors of their chondroitin sulfate bound-heavy chains for transfer to hyaluronan in the presence of TSG6 [105–108]. This results in a crosslinked extracellular hyaluronan network [109] that is permissive for leukocyte cell adhesion [110,111]. Generation of such a matrix in joint tissues is likely involved in the acute phase of this model, since RNASeq data show a 1.8 +/- 0.12 and 2.7 +/- 0.11 fold increase in mRNA levels for TSG6 (*Tnfaip6*) and hyaluronan synthase 2 (*Has2*), respectively at 3d post CFA, with both genes returning to pre-inflammation levels by 7d. Moreover, following the acute inflammatory response, the perimeniscal synovium was transformed into a hyperplastic tissue with a fibrotic matrix that was extensively populated by myeloid cells, but no significant presence of T-lymphocytes was detected.

Recent studies of RA patient-derived synovial biopsies have identified subgroups of synovial "pathotypes" [112,113]. Interestingly, our data suggest that CFA-IAI results in formation of the mixed myeloid/fibroid, but not the lymphoid pathotype. The former is reported to be treatment-resistant to many current DMARDs contributing to persistent "low joint disease activity" of many RA patients. This underscores the utility of this model to develop novel therapeutic approaches targeting joint tissue destruction in inflammatory arthritis. The development of a macrophage-rich synovial tissue is supported by the RNASeq based pathway analyses. Firstly, post-CFA activation of signaling pathways, including Notch, JAK/STAT, NF-κB, and MAPK [114], metabolic reprogramming, and increased glycolytic pathways are known to control macrophage polarization towards the pro-inflammatory phenotype in RA. In addition, biosynthetic pathways for arginine, a substrate for the inducible nitric oxide synthase (iNOS) in macrophages, indicates elevated production of nitric oxide (NO) and citrulline, both of which are key metabolites in the inflammatory pathogenesis of RA [115,116]. Notably, persistent activation of the "lipolysis in adipocytes", fatty acid metabolism, as well as cortisol synthesis and secretion pathways seen in our studies have also been reported to be a markers for a highly pro-inflammatory synovial subtype in OA and RA patients [117–119]; and is consistent with turnover of the sub-synovial adipose tissues induced by joint inflammation.

The development of a pro-inflammatory macrophage-rich synovium is also likely responsible for epiphyseal and metaphyseal periosteal surface pitting present at 3d and 7d post-CFA, in regions that were invaded by the hyper-proliferative pannus [120]. A transient downregulation was present of propanoate, a short-chain fatty acid known for its role in anti-inflammatory macrophage polarization, and specifically in suppression of osteoclastic activities in particle-induced aseptic osteolysis [121–123]. Re-establishment of a smooth periosteal surface by 14d supports the conclusion that the specific macrophage-mediated inflammatory joint tissue responses in the CFA-model are self-limiting, which is further supported by a robust increase in transcripts for oxidative phosphorylation [124]. This pathway plays a critical role during the polarization of the macrophage phenotype from a pro-inflammatory, glycolysis-dependent type to the anti-inflammatory type [125,126]. This together with a mineralization response seen in the post-CFA reactive periosteum would account for the reversal of periosteal surface damage. On the other hand, trabecular bone recovery did not occur in keeping with published data [127], as remodeling cycles in this compartment are long compared to the 14d time frame of our experiment with timing of osteoclast recruitment to this site, lifespan of osteoclasts, and their lifespan in rats is on the order of months [128].

On the other hand, there was no attenuation of the increases in cytokine and chemokine signaling pathways or activated T and B cell characteristic pathways, suggesting that even though several inflammatory responses in the joint are transient, many of the CFA-induced cellular and metabolic changes are in keeping with an irreversible modification of the synovial tissues. Moreover, additional challenges to such joints could lead to more severe innate responses and eventually lead to activation of the adaptive immune system and development of a chronic inflammatory arthritis. It should also be noted that only a few DEGs during intestinal responses to CFA-IAI overlapped with those detected for the synovium. We show that *Nos2, S100a9, Cxcl1, Ccl2, and Lcn2* increased in the synovium and decreased in the intestine, whereas *C1ql3* and *Serpine1a* were decreased in the synovium and increased in the intestine.

Gross evaluation by histological assessment of the ileum and colon revealed no detectable changes in the overall appearance of the tissue during the 2-week post-CFA period. This included intactness of intestinal villi, no increases in vascularization of the submucosa, and unchanged cell proliferation indices, as determined by distribution and abundance of Ki67+cells. There were no marked changes in the abundance of goblet cells throughout the ileum and colon, with only a transient change in mucus secretion in the distal ileum. In addition, the microbiota composition was minimally affected, with only a mild and transient response during peak joint inflammation (7d post-CFA-IAI) and normalized by the 14d time point. Moreover, using immunohistochemistry, we detected only transient alterations in immune cell abundance, including increased CD68+macrophages in crypts of the colon and increased CD8+T cells at the apical surface of villi in the ileum. Moreover, such changes occurred in sporadic regions of the intestinal compartments. Effects of the above responses on intestinal functionality in the CFA model will need additional evaluations including intestinal permeability assessment (21)

and metabolomics studies [129]. In addition, specific immune cell associated pathways (Fig 9) in all three regions of the intestine, showed suppression of a broad range of immune-associated responses characteristic of resident leucocytes and dendritic cells, whereas lymphoid cell types were essentially unaffected in response to the joint inflammation. Such broad "gene silencing" might be expected as a response to prevent chronic damage to the essential functions of the intestine, including nutrient digestion and absorption, as well as a steady-state defense against bacterial invasion to support microbial symbiosis that might be affected if an innate immune response is activated in the gut by inflammation in a "distant" organ

Our data contrast with the widespread intestinal manifestations observed in the chronic rodent models of inflammatory arthritis induced by immunization with type 2 collagen [130]. In fact, our bulk RNASeq data analyses from the proximal and distal ileum and the distal colon indicate a widespread "allostatic" response [39] to the CFA-induced inflammation and pain stressors in the joint. We observed significant suppression of transcriptomes compared to levels seen in those tissues pretreatment and included downregulation of genes in multiple pathways controlling pro-inflammatory signaling, immune activation, and cellular metabolism. Such broad "gene silencing" might be expected as a response to prevent chronic damage to the essential functions of the intestine, including nutrient digestion and absorption, as well as defense against bacterial invasion and support microbial symbiosis. Our data showed containment of cytokine and chemokine signaling and inhibition of the oxidative phosphorylation (OXPHOS) and PPAR-γ pathways throughout the post-CFA-IAI period. OXPHOS and mitochondrial function are fundamental for metabolic homeostasis of all cell types of the intestine and imperative to a functional barrier, however, enhanced flux through this pathway occurs when increased ATP is needed by immune cell during their activation and cytokine production. Furthermore, mitigation of reactive oxygen species (ROS), generated as by-products of enhanced OXPHOS, would prevent their damaging effect on intestinal function [131], including disrupted barrier function, dysbiosis, and impaired immune responses [132,133].

Genes in a second pleiotropic signaling pathway, the PPAR-γ pathway, extensively studied in colon cancers [134] were also downregulated in the post-CFA-IAI period, and this occurred both in the intestinal tissues and the synovium. In the context of the latter, this activity is pivotal in regulation of adipose tissue metabolisms, and its downregulation is a hallmark of inflammatory responses [135], which is consistent with loss of adipocytes in the myeloid/fibrotic induced synovium from CFA-IAI. A protective effect of its downregulation in the intestinal tissues might be attributed to silencing its cross-signaling with a range of other signaling pathways, including NFκB, Notch [136], JAK/STAT [137,138], MAPK [139], and Wtn/β-catenin [135]; some of which were also found to be downregulated in the post-CFA intestines. In summary, we provide, for the first time, data on temporally coordinated cellular responses between joint tissues and the intestine following an acute inflammatory insult by ICFA-IAI into the knee joints of skeletally mature rats. Additional application of spatial transcriptomics to depict anatomical location of transcriptomic alterations in the intestinal barrier, together with ATACSeq [140] to identify regions of the genome that are accessible or blocked would further clarify regulatory mechanisms that could serve as therapeutic targets locally or systemically, during inter-organ communication in chronic arthritis. Moreover, additional detailed characterization of the myeloid cells (M1, M2 macophages) accumulated in synovium and intestinal lining will be included in future experimentation to delineate mechanism of synovial pathotype in eliciting an intestinal response.

Transmission of signals between diseased tissues such as bone, kidney [141], heart [142], liver [143] and brain [144] and the gut has been reported for a wide range of disease conditions, but the molecular and cell biological mechanisms remain largely undefined, except for the much-studied bidirectional pathway between the brain and the intestine [144]. The latter includes cytokine, chemokines, migrating or circulating immune cells, and microbiota-derived metabolites. More relevant as a potential mediator of intestinal responses to joint inflammation is the enteric nervous system (ENS), which is a highly conserved network of neuron and glial cells located throughout the intestinal [145]. In this study, it likely was indicated in the development of gut inflammation and dysbiosis in response to psychological stressors [146], such as pain-induced anxiety [147], and could also underlie the presence of the allostatic tissue response [148]. Indeed, our preliminary data suggest a likely neurobiological involvement in the intestinal response to joint inflammation induced by CFA-IAI.

Additional mining of our transcriptomic data sets indicated modulation of serotonergic pathways, glutamatergic synapse activity, synaptic vesicle recycling, and retrograde endocannabinoid signaling throughout both ileum and colon. Select transcripts of genes with known neurobiological effects, such *Gdf15*, *Atp1a3*, and *Mdga2* [149] were activated in the ileum [149–151], suggesting that in future mechanistic studies for joint-gut communications, potential Vagus/ENS responses should be assessed. Although systemic inflammatory auto-antibodies or T and B cell responses have not been detected in IA CFA rodent models, locally mediated humoral reactions might also contribute to the inter-organ responses seen here.

In conclusion, since the CFA-induced joint inflammation model does not develop chronic systemic inflammatory conditions, it remains to be established whether multiple or chronic or consecutive allostatic responses of the intestinal tissue to joint inflammation will result in 'allostatic overload' [152]. Such conditions might ultimately lead to pathological disruption of the intestinal barrier function with dysbiosis as reported in both inflammatory bowel disease [153] and might also underlie the intestinal dysfunction reported in both patients and animal models of inflammatory arthritis [2,21].

## Methods

*Animal husbandry*. All procedures and training of research staff for this project were approved and in compliance with the Institutional Animal Care and Use Committee (IACUC) at Rush University Medical Center (Protocol 21–077). A total of 44 male Sprague Dawley (SD) rats (375-400g, approximately 4 months old, Envigo) were pair-housed in cages (polysulfone,19 × 10.5 × 8 in.) in the animal facility maintained at 22°C and 35%−55% humidity with a 12-hour light/dark cycle (light 7 AM-7 PM). All rats were provided with nesting material (Cob + Plus bedding with alpha twist). Standard laboratory-grade rodent chow (Teklad Global Rodent Diet 2018, 18% protein; Envigo) and water were provided *ad libitum* (S1 Fig). All rats were monitored every 3 days for animal welfare. There was no suffering or distress observed, and therefore no analgesics or anesthetics were administered outside of IAI and blood collection procedures.

*Induction of knee joint inflammation with Complete Freund's adjuvant*. Experiments were initiated after a 2-week acclimatization period to minimize effects of stress conditions during transport and housing, severity, and inter-group inconsistency in disease responses. Intra-articular injections (IAI), blood collections and sacrifice were all conducted at the same time of day, i.e., between 9AM-noon. Rats were assigned to one of three study groups (n = 6 rats/group): Naïve (no intra-articular injection), Complete Freund's Adjuvant (CFA, InvivoGen, VacciGrade, vac-cfa-10) 10 mg Mycobacterium tuberculosis, 1.5 ml Mannide monooleate and 8.5 ml paraffin oil, InvivoGen) and vehicle controls (1xPBS). Under 3% isoflurane anesthesia, IAI was administered using a 29G insulin syringe lateral to the patellar tendon as a single 50µl dose bilaterally, into the left and right knee joints. Also under 3% isoflurane anesthesia, rats were weighed, and knee and ankle diameters measured using calipers in the medial-lateral orientation on day 0 (day before IAI), 3, 7 and 14 post-IAI. No rats died during the experimental timeframe. Non-fasted tail-vein blood was collected into a BD Vacutainer SST tube per rat. Blood was allowed to clot at room temperature. Serum was separated by centrifugation at 3,000 rpm for 18 min at 4°C (~250 µL per bleed) and 40 µL 10X proteinase inhibitor (PI) cocktail (cOmplete, Mini, EDTA-free Protease Inhibitor Cocktail, Roche, 11836170001) was added prior to storage at −80°C.

*Sacrifice, tissue collections, and sample storage* Rats were euthanized longitudinally by $CO_2$ inhalation and secondary cervical dislocation at 3, 7 or 14 days post-IAI (S1B Fig). Sample size per outcome measure is also listed in S1C Fig. At sacrifice, bilateral hind limbs were collected from all rats and disarticulated at the hip. From right hind limbs, synovial fluid was collected by opening the joint capsule while the knee was carefully flexed. To retrieve synovial fluid the joint cavity was rinsed twice with 150 µL 1xPBS and collected into sterile microcentrifuge tubes. Lavages from right and left knee joints from each individual rat were combined and centrifuged at 3,000 rpm/805 g for 10 min at 4°C to remove cells and debris. Supernatants were stored at −80°C until further analyses. To minimize difficulties in result interpretation due to potential sampling errors of synovium from the rat knee joint, synovial tissues were collected from the readily identifiable peri-meniscal sites and the patellar fat pad, placed into RNALater and stored at −80°C for RNA-Seq analyses. From the right disarticulated hind limbs, femora and tibiae were fixed in 10%NBF for 7 days and then moved to 70% EtOH until they

underwent micro-computed tomography scanning (Scanco 50, Switzerland). Left intact knee joints were isolated by making transverse cuts 1.5 cm proximal and distal to the joint and then fixed in 10% neutral-buffered formalin (NBF). Joints were decalcified in 20% EDTA, cut at midline sagittally and paraffin processed, embedded and sectioned at 6μm slices for histology. Right intact knee joints were dissected open using a fresh scalpel. Synovium and patellar fat pad were excised into RNALater (Invitrogen) and froze at −80°C until RNA extraction. Ileum and colon were dissected and fecal contents expressed into sterile 1.5 ml microcentrifuge tubes from both gut segments and stored at −80°C for 16S microbiome analyses. The remaining tissues were either rinsed in ice-cold 1xPBS, transferred to RNALater and stored at −80°C for RNA-Seq, or placed into Modified Bouin's fixative and prepared into Swiss rolls [154] for paraffin embedding, and sectioned at 6μm thickness for histology and immunohistochemistry (S1 Fig).

*Histological and immunohistochemical evaluation of joint and intestinal tissues.* Sagittal sections from knee joints were stained with Hematoxylin/Eosin, anti-CD68 (Abcam, Ab283654; 1:100) or anti-CD4 Abcam, Ab237722 1:2000). Sections of intestinal Swiss rolls were stained using the Alcian Blue Kit (Vector Labs, H-3501), anti-CD68 (1:100), anti-CD4 (1:2000), anti-CD8 (Abcam, Ab237709, 1:500) or anti-Ki67 (Invitrogen, MA5–14520; 1:100), Invitrogen, MA5–14520) or the following biotinylated lectins (all from Vector Labs), *Griffonia Bandeiraea Simplidifolia* Lectin I, (GSL I, B-1105–2), *Maackia Amurensis Lectin II* (MAL II, B-1265–1) and *Ulex Europaeus Agglutinin I* (UEA, B-1065–2). Prior to histochemical staining, antigen retrieval was performed using Tris/EDTA (Abcam, ab93684), and sections were blocked with normal goat serum (Vector Laboratories, S-1000). Immuno- and lectin reactivities were developed using the ABC kit (Vector Laboratories, PK-7100) and the DAB kit (Vector Laboratories, SK-4100). Methyl green (Vector Laboratories, H-3402) was used as a counterstain. Slides were imaged using an Olympus VS200 Scanner.

*Western blot characterization of affinity-purified antibodies to mouse bikunin.* A synthetic peptide of mouse bikunin was synthesized and conjugated to Keyhole Limpet Hemocyanin (KLH), via a non-authentic cysteine residue at the C-terminal end, for production of a polyclonal antibody in a rabbit (Mimotopes, Australia). The anti-serum (from rabbit A) was affinity purified using the immunizing peptide (Genscript, USA) covalently coupled via the cysteine residue to SulphoLink Coupling Resin (Thermo Scientific Pierce, UK), according to the manufacturer's instructions. Purified IgG fractions were pooled, concentrated and stored in aliquots at −20°C; the affinity purified antibody is designated ap_A_mBikunin. A Western blot of mouse serum treated with/without chondrotinase ABC lyase or NaOH was used to characterize the affinity purified antibodies. Here 1.25 μL of mouse serum was incubated with/without 0.1 U Chondroitinase ABC lyase (from *Proteus vulgaris*; Sigma Aldrich) for 2 h at 37°C or 0.1 M NaOH for 10 min at RT after which 1.0 μL 1 M HCl was added to the NaOH-treated samples. Reactions were stopped by adding 5 μl of 5X sample loading buffer; 12 μl of each reaction was run on a NOVEX 4–20% Tris-Glycine gel. Proteins were transferred to a nitrocellulose membrane in Tris-Glycine transfer buffer. Blots were blocked for 1 h with 5% (w/v) dried milk in PBS/0.05% (v/v) Tween-20 (PBST), washed briefly in PBST and then incubated with ap_A_mBikunin (diluted 1:500 in PBST with 5% (w/v) dried milk) overnight at 4°C. After washing in PBST the membrane was incubated with IRDye 800CW-conjugated goat anti-rabbit IgG secondary antibody (1:5000 in PBS; LI-COR Biosciences Ltd) for 1 h at room temperature, washed as before and visualized using an Odyssey CLx imaging system (LI-COR). The species detected by the antibody are indicated to the right in S2 Fig.

*Western blot analyses of serum and synovial fluid bikunin.* Commercially available mouse serum, which was processed the same as experimental rat serum, was included in each western blot as an internal standard. Serum samples were thawed on ice and diluted 1:1 with 50mM ammonium acetate pH 7.0 containing 1XPI and debris removed with a 0.45μM filters. 150 μL of sera were desalted using MicroCon 3 filtration units (at 12,500 rpm in a microfuge for 15 min) and retentates washed once with 450 μL 50mM ammonium acetate, pH 7.0. Desalted proteins were collected from the membrane with 500 μL ammonium acetate, pH 7.0. Portions (containing 7.5 μL equivalents of serum) were incubated at 37°C for 2 hrs. with or without 5mU of protease-free chondroitinase ABC lyase (Chondroitinase ABC protease free).

Synovial fluid samples were thawed on ice, diluted to 450 μL with 50mM ammonium acetate, pH 7.0 containing 1XPI, clarified by centrifugation at 12,000 g at 4°C for 15 min, and desalted using MicroCon 3 filtration units as described above

for serum samples. The retentates were recovered in 200 μL 50mM ammonium acetate, pH 7.0 and 100 μL portions were incubated at 37°C for 2h with or without 5mU of protease-free chondroitinase ABC.

Sera and synovial fluid samples were speedvac dried and analyzed by gel electrophoresis and western blot. Chondroitinase digestion buffer was removed by speedvac evaporation. Samples were dissolved in Tris-glycine SDS sample buffer (Novex, AMS.E1028-02) containing 100μl of 0.5mM DTT, heated at 90°C for 5min before electrophoresis on SDS gels (Invitrogen Novex WedgeWell 4–12% Tris-Glycine). Separated proteins were electro-transferred on ice (250V, 302μA for 45min and 150V, 297μA for 15min) to 0.45μm nitrocellulose membranes (BioRad) and these were incubated with affinity purified anti-bikunin antibody (ap_A_mBikunin). Antibody reactivity with inter-α-inhibitor (IαI) and pre-α-inhibitor (PαI), bikunin, and bikunin core protein [106] was characterized (S2A Fig). The Western blotting procedure (in S2B,C Fig) was performed as previously published [155]. Chemiluminescent images were recorded using iBright1500 Instrumentation and further analyzed using ImageJ software. Data are shown as integrated pixel density (IPD) relative to bikunin present in standard mouse serum (Sigma Aldrich m595).

*Micro-computed tomography analyses of femurs*. Micro CT scanning was completed using a Scanco μCT50 (Switzerland). Right femurs were scanned using 70kVp, 114μA, 500ms integration time, and 10μm voxel size. The region of interest (ROI) for trabecular and cortical bone analyses started just proximal to the distal femoral growth plate and continued 2mm proximally into the metaphysis. The trabecular and cortical bone compartments were analyzed separately using thresholds of 200 and 1000, respectively. Cortical and trabecular bone parameters were measured using the manufacturer's software. Periosteal reaction was noted when present and analyzed separately from the cortex using the same threshold values.

*Bulk RNASeq analyses of synovium and intestine*. To obtain reproducible and high-quality yields of RNA from synovium, tissues were harvested from right and left knees and pooled from n = 3 rats then analyzed as a single sample. Distal ileum and colon tissues in RNAlater were thawed on ice and 1cm pieces were dissected out (including villi, submucosa, and smooth muscle layers) using transverse cuts. The number of intestinal tissue replicas to provide sufficient DEG detection power was determined as previously described [156]. All tissues were collected from sex and age matched rats.

Using previously published methods [157] RNA was extracted and purified from all tissues, samples with RIN numbers ≥7 were reverse transcribed into cDNA, PCR amplified with primers CS1_515F and CS2_806R followed by library preparation. Bioinformatic analyses of sequencing data was carried out as follows: Raw reads were trimmed to remove Truseq adapters and bases from the 3' end with quality scores less than 20 using cutadapt [158]; trimmed reads shorter than 40bp were discarded, and were then aligned to the mRatBN7.2 *Rattus norvegicus* (Norway rat) reference genome using STAR [159]. The expression level of ENSEMBL genes was quantified using featureCounts [160] and the resulting expression counts were used to compute differential expression statistics with edgeR [161, 162]. Normalized expression was computed as $\log_2$ counts per million (CPM), including a TMM normalization and batch effect correction. In all cases p-values were adjusted for multiple testing using the false discovery rate (FDR) correction of Benjamini and Hochberg [163]. Volcano plots were generated within the R programming language. DEG (Differentially Expressed Genes) with an absolute $\log_2$FC (Log Fold Change) <+2 or <−2 and a q value of less than 0.05 were assigned to specific pathways using the KEGG and further grouped under seven functional categories: Metabolism, Cellular Processes, Genetic Information Processing, Environmental Processing and Immune System

*16S microbiome analyses*. Details have been described by us previously [23]. Briefly, DNA was extracted from fecal samples using a Maxwell RSC48 device (Promega), with bead-beating implemented off-instrument. Microbial community structure was characterized by DNA extraction, PCR amplification and sequencing of 16S ribosomal RNA gene amplicons [164]. Microbiome bioinformatics was performed with QIIME2 2021.11 [165]. Raw sequence data were checked for quality using FastQC and merged using PEAR [166]. Merged sequences were quality filtered using the q2-demux plugin followed by denoising with DADA2 [167] (via q2-dada2). Primer adapter sequences were removed using cutadapt algorithm [158]. Alpha-diversity metrics (observed features [168], Shannon Index [169], Simpson's Index [170], and Pielou's Evenness) and beta diversity metrics were calculated using q2-diversity after samples were rarefied to a depth of 59,000 sequences

per sample. Taxonomy was assigned to ASVs using the q2-feature-classifier [171] classify-sklearn naïve Bayes taxonomy classifier against the SILVA 138 99% reference sequences database [172]. Differentially abundant genus level taxa between pairwise groups were identified using the compositional centered Log-ratio Kruskal Wallis (CLR-KW) algorithm. Adjusted P-values (also known as q-values) were generated using the Benjamini–Hochberg method [163].

*Statistical Analyses for Bone Data.* Control and test rats from 2 separate cohorts (one in the summer and one in the winter) were included. For the trabecular parameters, we observed a cohort effect so we express all test results normalized to the mean of the cohort-specific control animals. Significance was assessed using one-way analyses of variance (ANOVA) for group effects with Bonferroni post hoc tests. All results are reported as means ± standard deviation and individual data points are shown. The statistical testing was performed with commercially available software (SPSS v.19 for Windows, Chicago, IL).

## Supporting information

**S1 Fig. Overview of experimental protocol. (A)** Schematic of *in vivo* CFA Model (see Methods for Details). Rats received one intra-articular injection (IAI) of 50 µL of CFA (10 mg/ml) or sterile saline into both knee joints **(B).** Number of rats used for each study group (Naïve, Saline, CFA). Experiments were performed on 2 separate cohorts but procured from the same supplier. S3 = Proximal Ileum (PI), S4 = Distal Ileum (DI) C = Colon (Co). **(C)** Segmentation of small intestine and colon for RNASeq analyses and histological evaluations (see Methods for details). S = small intestine, S3 = proximal Ileum, S4 = distal Ileum, C = distal colon.
(PPTX)

**S2 Fig. Characterization of anti-bikunin antibody ap_A_mBikunin, Western blots analyses of bikunin complexes in rat sera and synovial fluid, and detection of serum proteins in knee joint synovial fluids post-IAI-CFA. (A)** A Western Blot of mouse serum treated with/without chondroitinase ABC lyase (Ch'ase) or NaOH to release bikunin or bikunin•CS, respectively from IαI, PαI and bikunin•HS species was used to characterize a rabbit anti-mouse bikunin antibody (denoted ap_A_mBikunin) as described in the Methods. Note that the mouse sequence of the immunizing peptide, AVLPQE**S**EGS, is highly similar to the corresponding rat sequence, AVLPQE**N**EGS, with only one amino acid difference. **(B)** Western blot of rat sera, collected from 4 individual rats prior to IAI-CFA, and mouse serum (as a standard) were prepared and treated without (lanes 1–4) or with (lanes 5–8) chondroitinase ABC lyase (Ch ABC) prior to electrophoresis as described in the Methods. **(C)** Western blot of rat sera of knee joint synovial fluid lavages collected from 3 individual rats at d3 post IAI-CFA and 'standard' mouse serum were prepared and treated or not with chondroitinase ABC (Ch ABC) prior to electrophoresis as described in the Methods. Positions of bikunin-contianing species in **B** and **C** (i.e., IαI, PαI, bikunin-CS and bikunin) are indicated to the right of the blots. All membranes (in **A**, **B**, **C**) were incubated with ap_A_mBikunin as described in the Methods. **(D)** To assess effusion of serum proteins into the joint space following IAI-CFA, portions of rat synovial fluid lavages collected prior to IAI-CFA (d0) and at d3, d7 and d14 post IAI-CFA, and rat serum pre-IAI-CFA, were electrophoresed with or without prior treatment with chondroitinase ABC (Ch ABC), and gels were Coomassie stained. Expected migration positions of lactoferrin, albumin and IgG are shown to the right.
(PPTX)

**S3 Fig. Heat map of DEGs in proximal ileum (A), distal ileum (B) and distal colon (C) modified only at a single time point post-CFA-IAI.**
(PPTX)

**S4 Fig. Heat map of DEGs in proximal ileum (A), distal ileum (B) and distal colon (C) modified at 2 or 3 time points post-CFA-IAI.**
(PPTX)

**S5 Fig. Dot plot representation of transcriptomic changes in synovial tissue following CFA-IAI.** Pathways (ranked by p-value) were assigned to color-coded functional categories as listed in the KEGG Database (Metabolism, Cellular Processes, Genetic Information Processing, Environmental Processing, Immune System, Nervous System). Gene ratio for a given pathway is computed as the percentage of genes present divided by the total number of genes in that pathway. Activated and Suppressed pathways are separated by a solid black vertical line and gene ratios <or > 0.5 are separated by a dotted vertical line. The top 20 modified pathways are above the horizontal dotted line.
(PPTX)

**S6 Fig. Dot plot representation of transcriptomic changes in synovial tissue following PBS-IAI.** Pathways (ranked by p-value) were assigned to color-coded functional categories as listed in the KEGG Database (Metabolism, Cellular Processes, Genetic Information Processing, Environmental Processing, Immune System, Nervous System). Gene ratio for a given pathway is computed as the percentage of genes present divided by the total number of genes in that pathway. Activated and Suppressed pathways are separated by a solid black vertical line and gene ratios <or > 0.5 are separated by a dotted vertical line. The top 20 modified pathways are above the horizontal dotted line.
(PPTX)

**S7 Fig. Dot plot representation of transcriptomic changes in the proximal ileum following CFA-IAI.** Pathways (ranked by p-value) were assigned to color-coded functional categories as listed in the KEGG Database (Metabolism, Cellular Processes, Genetic Information Processing, Environmental Processing, Immune System, Nervous System). Gene ratio for a given pathway is computed as the percentage of genes present divided by the total number of genes in that pathway. Activated and Suppressed pathways are separated by a solid black vertical line and gene ratios <or > 0.5 are separated by a dotted vertical line. The top 20 modified pathways are indicated by a dotted vertical line.
(PPTX)

**S8 Fig. Dot plot representation of transcriptomic changes in the distal ileum following CFA-IAI.** Activated pathways are listed on the left and suppressed pathways on the right. Gene ratios and pathway groupings are assigned as described for S6 Fig.
(PPTX)

**S9 Fig. Dot plot representation of transcriptomic changes in the colon following CFA-IAI.** Activated pathways are listed on the left and suppressed pathways on the right. Gene ratios and pathway groupings are assigned as described for S6 Fig.
(PPTX)

**S10 Fig. Immunohistochemical localization of CD68 + cells in proximal and distal regions of the ileum and colon.** Images represent regions from the scanned slides that were used for panels shown in Fig 10.
(PPTX)

**S11 Fig. Immunohistochemical localization of CD8 + cells in proximal and distal regions of the ileum and colon.** Images represent regions from the scanned slides that were used for panels shown in Fig 10.
(PPTX)

**S12 Fig. Immunohistochemical localization of CD4 + cells in proximal and distal regions of the ileum and colon.** Images represent regions from the scanned slides that were used for panels shown in Fig 10.
(PPTX)

**S13 Fig. Immunohistochemical localization of Ki67 + cells in proximal and distal regions of the ileum and colon.** Images represent regions from the scanned slides that were used for panels shown in Fig 10.
(PPTX)

**S14 Fig. *Lectin histochemistry in proximal and distal regions of the ileum and colon.*** Images represent regions from the scanned slides that were used for panels shown in Fig. 10. Mal II (A) and UEA (B).
(PPTX)

**S15 Fig. Abundance of mucin degrading bacteria in feces from distal ileum and distal colon pre- and post-CFA-IAI.** Raw counts for mucin-degrading bacteria (Akkermansia, Alistipes) and sialic acid cleaving bacteria (Bifidobacterium) are shown. Note that in pre-treatment rats, Akkermansia and Alistipes were below detection in the ileum (ND = not detected). Variable animal-to-animal responses are outlined in dotted lines. Note the much lower counts for *Akkermansia* and *Alistipes* in the ileum compared to the colon. Bifidobacterium showed higher counts in both intestinal regions.
(PPTX)

## Acknowledgments

We would like to thank Dr. Georgia Papavasiliou (Illinois Institute of Technology) and Dr. Joseph Reynold (Rosalind Franklin University) for their insightful discussions on intestinal immunology and therapeutic target discovery during the progression of this project; Dr. Stefan Green for his input in study design and genomic analyses; and Rylan Martin for technical support.

## Author contributions

**Conceptualization:** Meghan M. Moran, Anthony J. Day, D. Rick Sumner, Anna Plaas.

**Data curation:** Meghan M. Moran, Jun Li, Quan Shen, Sheona P. Drummond, Caroline M Milner, Anthony J. Day, Ankur Naqib, Anna Plaas.

**Formal analysis:** Meghan M. Moran, Jun Li, Quan Shen, Sheona P. Drummond, Caroline M Milner, Anthony J. Day, Ankur Naqib, Anna Plaas.

**Funding acquisition:** Meghan M. Moran, Caroline M Milner, Anthony J. Day, Anna Plaas.

**Investigation:** Meghan M. Moran, Caroline M Milner, Anna Plaas.

**Methodology:** Meghan M. Moran, Jun Li, Caroline M Milner, Anthony J. Day, Ankur Naqib, Anna Plaas.

**Project administration:** Meghan M. Moran, Anna Plaas.

**Resources:** Meghan M. Moran.

**Software:** Ankur Naqib.

**Supervision:** Meghan M. Moran, Caroline M Milner, Anthony J. Day, D. Rick Sumner, Anna Plaas.

**Validation:** Sheona P. Drummond, Caroline M Milner, Anthony J. Day, Ankur Naqib, Anna Plaas.

**Visualization:** Meghan M. Moran, Ankur Naqib, D. Rick Sumner, Anna Plaas.

**Writing – original draft:** Meghan M. Moran, Caroline M Milner, Anthony J. Day, Ankur Naqib, D. Rick Sumner, Anna Plaas.

**Writing – review & editing:** Meghan M. Moran, Jun Li, Quan Shen, Sheona P. Drummond, Caroline M Milner, Anthony J. Day, Ankur Naqib, D. Rick Sumner, Anna Plaas.

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
