## [Decision Letter · Decision Letter 0]

3 Sep 2025

Dear Dr. Moran,

Thank you for submitting your manuscript to PLOS ONE. After careful consideration, we feel that it has merit but does not fully meet PLOS ONE’s publication criteria as it currently stands. Therefore, we invite you to submit a revised version of the manuscript that addresses the points raised during the review process.

**ACADEMIC EDITOR:** Thank you for submitting your manuscript to PLOS ONE. Both reviewers indicate that your study has merit but have provided constructive suggestions for improvement of your work. Therefore, we invite you to submit a revised version of the manuscript that comprehensively addresses all points raised during the review process.

We look forward to receiving your revised manuscript.

Kind regards,

Andre van Wijnen

Academic Editor

PLOS ONE

Journal Requirements:

1. Please ensure that your manuscript meets PLOS ONE's style requirements, including those for file naming. The PLOS ONE style templates can be found at https://journals.plos.org/plosone/s/file?id=wjVg/PLOSOne_formatting_sample_main_body.pdf and https://journals.plos.org/plosone/s/file?id=ba62/PLOSOne_formatting_sample_title_authors_affiliations.pdf.

[We would like to thank Dr. Georgia Papavasiliou (Illinois Institute of Technology) and Dr. Joseph Reynold (Rosalind Franklin University) for their insightful discussions on intestinal immunology and therapeutic target discovery during the progression of this project; Dr. Stefan Green for his input in study design and genomic analyses; and Rylan Martin for technical support. Funding for this project is from the Orthopedics Departmental Fund RUMC (MMM), the Katz/Rubschlager Endowed Chair at RUMC (AP), and Versus Arthritis, UK Grant 22277 (AJD and CMM).]

[Funding for this project is from the Orthopedics Departmental Fund RUMC, no grant number (MMM), the Katz/Rubschlager Endowed Chair at RUMC (AP), and Versus Arthritis, UK Grant 22277 (AJD and CMM). None of the sponsors or funders had any role in the study design, data collection and analysis, decision to publish, or preparation of the manuscript.]

8. We notice that your supplementary figures are uploaded with the file type 'Figure'. Please amend the file type to 'Supporting Information'. Please ensure that each Supporting Information file has a legend listed in the manuscript after the references list.

Reviewers' comments:

Reviewer's Responses to Questions

**Comments to the Author**

1. Is the manuscript technically sound, and do the data support the conclusions?

Reviewer #1: Yes

Reviewer #2: Yes

2. Has the statistical analysis been performed appropriately and rigorously?

Reviewer #1: Yes

Reviewer #2: Yes

3. Have the authors made all data underlying the findings in their manuscript fully available?

Reviewer #1: Yes

Reviewer #2: Yes

4. Is the manuscript presented in an intelligible fashion and written in standard English?

Reviewer #1: Yes

Reviewer #2: Yes

Reviewer #1: The manuscript by Meghan Moran and colleagues entitled "Evidence of an Allostatic Response by Intestinal Tissues Following Induction of Joint Inflammation", presents an interesting and well-conducted study. While the writing is generally good, certain sections would benefit from improved cohesion and clarity. My minor comments are as follows:

1. Include relevant statistical data on the disease burden of gut health issues linked to arthritis, bone disorders, or the joint–gut communication axis.

2. Consider adding, either in the introduction or discussion, how metabolic-inflammatory disorders are associated with arthritis—particularly osteoarthritis as shown in recent studies (PMID: 34537381).

3. In the discussion: the statement "Transmission of signals between diseased tissues and the gut", clarify what is meant by diseased tissues.

4. For methods section, provide detailed catalogue numbers, clone numbers for antibodies, and company names for all kits and reagents used.

5. Clarify whether the bulk RNA-seq analyses of synovium and intestine were generated from age-matched animals.

6. Expand on the synovial fluid collection procedure.

7. Add missing p-values to most of the statistical data in Figure 1 panels.

8. For Figures 4 to 6, consider including IPA or GO pathway analyses to identify cell-specific immune signaling pathways for myeloid and lymphoid populations (not just broad terminology) alongside the volcano plots in the same figure panel

Reviewer #2: The manuscript “Evidence of an Allostatic Response by Intestinal Tissues Following Induction of Joint Inflammation” investigates the inter-organ communication between inflamed joint tissues and the intestinal barrier using a rat model of mono-joint inflammatory arthritis induced by Complete Freund’s Adjuvant (CFA). The authors explore how acute joint inflammation affects intestinal tissues at the molecular, cellular, and microbiome levels, proposing an allostatic response—a concept from stress physiology—within the gut.

The text is well writing and clear, so I have few concerns about it, as explicated below:

The title brings the term joint inflammation, but the central topic of the manuscript is an experimental rheumatoid arthritis. Can the authors explain the choice of the term in the title?

In the phrase “However, it remains to be determined if the intestinal responses are mediated by a disrupted systemic immune response in RA, and/or if the bone and cartilage destruction mediated by an inflamed synovium (“RA pannus”) is the central link driving the intestinal pathogenesis in RA.” the authors are trying to suggest that systemic or localized immune response contribute to RA pathogenesis? It is not clear...

The fluid of joints was recovered using PBS in a rinse. Can this procedure dilute the collected fluid? Besides, the manuscript does not clearly specify the volume of synovial fluid collected, which is crucial for interpreting protein concentration data.

What is the abbreviation FFPE?

Please, indicate where is possible, the “n” of animals used in each experiment, mainly in the figure legends.

CD68 is a general marker of macrophages. Did the authors could illustrate these cells with a more specific marker to define them as M1 or M2 macrophage. The same can be imagined to T cells (Th1 or Th2).

I think the correct in the first paragraph of page 12 is Fig. 11C.

In the phrase “Our data contrast with the widespread intestinal manifestations observed in the chronic rodent models of inflammatory arthritis induced by immunization with type 2 collagen” the authors discuss the difference of results between this study and literature, emphasizing the models of chronic inflammations used by other studies. Could not the authors use references about experimental RA? Besides, the authors explain in the text that the model of RA used in this study does not mimic literally a chronic manifestation.

The study is described as a mono-joint model, yet the systemic implications (e.g., gut responses) are emphasized. This raises questions about whether the observed intestinal changes are truly systemic or locally mediated via neural or humoral pathways.

While the authors claim minimal changes in microbiome composition, the analysis appears superficial. More detailed taxonomic and functional profiling would strengthen this claim.

The study relies heavily on transcriptomic data. Functional assays (e.g., permeability tests, cytokine quantification) would validate the proposed allostatic mechanisms.

Finally, a better clarification in the discussion section about the response of the gut to systemic inflammation caused by RA or to localized joint signals will enrich the manuscript.

**Do you want your identity to be public for this peer review?** For information about this choice, including consent withdrawal, please see our Privacy Policy

Reviewer #1: No

Reviewer #2: No

---

## [Author Response · Author response to Decision Letter 1]

27 Oct 2025

Below are our responses to reviewers. This was uploaded as a word document as well. Thank you.

Journal Requirements:

1. Please ensure that your manuscript meets PLOS ONE's style requirements, including those for file naming. The PLOS ONE style templates can be found at https://journals.plos.org/plosone/s/file?id=wjVg/PLOSOne_formatting_sample_main_body.pdf and https://journals.plos.org/plosone/s/file?id=ba62/PLOSOne_formatting_sample_title_authors_affiliations.pdf.

- Response: The blots shown in the Supporting Information Figure S2 contain full, unadjusted and uncropped gels.

[We would like to thank Dr. Georgia Papavasiliou (Illinois Institute of Technology) and Dr. Joseph Reynold (Rosalind Franklin University) for their insightful discussions on intestinal immunology and therapeutic target discovery during the progression of this project; Dr. Stefan Green for his input in study design and genomic analyses; and Rylan Martin for technical support. Funding for this project is from the Orthopedics Departmental Fund RUMC (MMM), the Katz/Rubschlager Endowed Chair at RUMC (AP), and Versus Arthritis, UK Grant 22277 (AJD and CMM).]

We note that you have provided funding information that is not currently declared in your Funding Statement. However, funding information should not appear in the Acknowledgments section or other areas of your manuscript. We will only publish funding information present in the Funding Statement section of the online submission form. Please remove any funding-related text from the manuscript and let us know how you would like to update your Funding Statement. Currently, your Funding Statement reads as follows: [Funding for this project is from the Orthopedics Departmental Fund RUMC, no grant number (MMM), the Katz/Rubschlager Endowed Chair at RUMC (AP), and Versus Arthritis, UK Grant 22277 (AJD and CMM). None of the sponsors or funders had any role in the study design, data collection and analysis, decision to publish, or preparation of the manuscript.]

- Response: Thanks- we have removed the funding information from the acknowledgements and added this text to our cover letter for it to be added by PlosOne to the appropriate form.

- Response: Thank you for this reminder. All sequencing reads generated in this study have been deposited in the National Center for Biotechnology Information (NCBI) BioProject database under accession numbers PRJNA 1347841 and 1348463 (RNASeq), PRJNA 1348465 and 1348470 (Microbiota). This statement has been added to the data sharing and availability statement.

5. We note that you have included the phrase “data not shown” in your manuscript. Unfortunately, this does not meet our data sharing requirements. PLOS does not permit references to inaccessible data. We require that authors provide all relevant data within the paper, Supporting Information files, or in an acceptable, public repository. Please add a citation to support this phrase or upload the data that corresponds with these findings to a stable repository (such as Figshare or Dryad) and provide URLs, DOIs, or accession numbers that may be used to access these data. Or, if the data are not a core part of the research being presented in your study, we ask that you remove the phrase that refers to these data.

- Response: Thank you for this comment. We have removed the ‘data not shown’ and added the fold values +/- SD to the Discussion text pg. 14: “Generation of such a matrix in joint tissues is likely involved in the acute phase of this model, since RNASeq data show a 1.8 +/- 0.12 and 2.7 +/- 0.11 fold increase in mRNA levels for TSG6 (Tnfaip6) and hyaluronan synthase 2 (Has2), respectively at 3d post CFA, with both genes returning to pre-inflammation levels by 7d”.

- Response: Our study uses a preclinical model and not human subjects or tissues therefore no IRB is required to be reported. We have included ethical treatment of animals protocol number to the methods section.

- Response: We have added the supporting information figure legend at the end of the manuscript.

8. We notice that your supplementary figures are uploaded with the file type 'Figure'. Please amend the file type to 'Supporting Information'. Please ensure that each Supporting Information file has a legend listed in the manuscript after the references list.

- Response: Thank you for this comment. We have added the supporting information figure legend at the end of the manuscript and uploaded the supplementary figures as Supporting Information Figures.

- Response: Thank you. We have included citations as suggested by reviewers, now cited in the text and added to the reference list.

- We thank the reviewers for their careful reading of the manuscript and their helpful comments. We have addressed each of the queries below and modifications are shown in the manuscript using track changes.

5. Review Comments to the Author

Reviewer #1: The manuscript by Meghan Moran and colleagues entitled "Evidence of an Allostatic Response by Intestinal Tissues Following Induction of Joint Inflammation", presents an interesting and well-conducted study. While the writing is generally good, certain sections would benefit from improved cohesion and clarity. My minor comments are as follows:

Query 1. Include relevant statistical data on the disease burden of gut health issues linked to arthritis, bone disorders, or the joint–gut communication axis.

Response: Thanks for this comment. We have edited and added new text and additional citations that now highlight statistical data on GI symptoms in RA and other bone diseases in the Introduction, 1st paragraph.

Query 2. Consider adding, either in the introduction or discussion, how metabolic-inflammatory disorders are associated with arthritis—particularly osteoarthritis as shown in recent studies (PMID: 34537381).

Response: Thank you for drawing attention to this reference. Whilst it does not directly address the effect of joint inflammation on intestinal health, we have added it in the Introduction Page 3: (REF PMID: 34537381) to draw attention to the possible effect of metabolic disorders on progression of OA due their effect on gut health.

Query 3. In the discussion: the statement "Transmission of signals between diseased tissues and the gut", clarify what is meant by diseased tissues.

Response: We have modified the text on Page 16 in the Discussion and added pertinent new references: “ Transmission of signals between diseased tissues such as bone, kidney (PMID: 31560235), heart (PMID: 40403109), liver (PMID: 38280375) and brain (PMID: 34822299) and the gut has been reported for a wide range of disease conditions, but the molecular and cell biological mechanisms remain largely undefined, except for the much-studied bidirectional pathway between the brain and the intestine [142], The later includes cytokine, chemokines, migrating or circulating immune cells, and microbiota-derived metabolites. More relevant as a potential mediator of intestinal responses to joint inflammation is the enteric nervous system (ENS), which is a highly conserved network of neuron and glial cells located throughout the intestinal tract [143].”

Query 4. For methods section, provide detailed catalogue numbers, clone numbers for antibodies, and company names for all kits and reagents used.

Response: Thank you for this comment. We have added the company and catalogue information to the Methods Section, pg. 18.

Query 5. Clarify whether the bulk RNA-seq analyses of synovium and intestine were generated from age-matched animals.

Response: We have added a statement to the Method Section Page 19: “All tissues were collected from sex and age matched rats”.

Query 6. Expand on the synovial fluid collection procedure.

Response: We have clarified the method description on page 13: “To retrieve synovial fluid the joint cavity was rinsed twice with 150µL 1xPBS and collected into sterile microcentrifuge tubes. Lavages from right and left knee joints from each individual rat were combined and centrifuged at 3,000 rpm/805 g for 10 min at 4°C to remove cells and debris. Supernatants were stored at -80ºC until further analyses.”

Query 7. Add missing p-values to most of the statistical data in Figure 1 panels.

Response: Thank you for this comment. The p-values are now represented in the graphs of Fig. 1 using * and a key that is in the legend.

Query 8. For Figures 4 to 6, consider including IPA or GO pathway analyses to identify cell-specific immune signaling pathways for myeloid and lymphoid populations (not just broad terminology) alongside the volcano plots in the same figure panel ---

Response: Thank you for this comment. For pathway analysis, we did indeed run GSEA against both KEGG database and the msigDB database. We used the results from KEGG in the manuscript as they are curated and named in a more understandable terminology. We do have results of pathway analysis GSEA against the msigDB Gene Ontology gene sets (subsets BP, MF, and CC). We added an additional table/figure view all the pathways and genes that were detected within the GO gene sets. We have added this as NEW FIGURE 9 with the Figure Legend “New Figure 9: Myeloid and Lymphoid-associated pathway analyses using GSEA using against the msigDB Gene Ontology gene sets, subset BP. Activated pathways are marked with a red upward pointing arrow (↑) and suppressed pathways with a blue downward pointing arrow (↓). No change is indicated by a horizontal double arrow (↔). SYN= synovium, PI = proximal ileum, DI = distal ileum, Co= Colon.”q

Also, the following text was added in the Results Section pg. 11: “Additional RNASeq data analyses to define modulation of Myeloid or Lymphoid-specific pathway using the GSEA data base against the msigDB Gene Ontology gene sets (subset BP) are shown in Fig 9. The majority of identified GOs, i.e. 22, were associated with cells from the myeloid lineage, 11 of those were for leukocytes and 5 for dendritic cells. All pathways, except for “Myeloid Development” in the distal colon, were downregulated relative to naïve intestinal regions. The number of affected pathways with decreased transcript abundance was largely in the proximal ileum, fewer in the distal colon, but largely unaffected in the distal ileum. Moreover, pathway modulations were most pronounced at 3d in all regions but largely restored to pre-injury levels by 14d post joint injury. Lymphoid pathways were minimally affected. “Lymphoid organ development” was suppressed in the proximal ileum and lymphoid progenitor cell differentiation activated at d14 in the distal colon.”

Lastly, we added text in the Discussion Section pg.15-16: “In addition, specific immune cell associated pathways (Fig. 9) in all three regions of the intestine, showed suppression of a broad range of immune-associated responses characteristic of resident leucocytes and dendritic cells, whereas lymphoid cell types were essentially unaffected in response to the joint inflammation. Such broad “gene silencing” might be expected as a response to prevent chronic damage to the essential functions of the intestine, including nutrient digestion and absorption, as well as a steady-state defense against bacterial invasion to support microbial symbiosis that might be affected if an innate immune response is activated in the gut by inflammation in a “distant” organ.”

Reviewer #2: The manuscript “Evidence of an Allostatic Response by Intestinal Tissues Following Induction of Joint Inflammation” investigates the inter-organ communication between inflamed joint tissues and the intestinal barrier using a rat model of mono-joint inflammatory arthritis induced by Complete Freund’s Adjuvant (CFA). The authors explore how acute joint inflammation affects intestinal tissues at the molecular, cellular, and microbiome levels, proposing an allostatic response—a concept from stress physiology—within the gut.

The text is well writing and clear, so I have few concerns about it, as explicated below:

Query 1: The title brings the term joint inflammation, but the central topic of the manuscript is an experimental rheumatoid arthritis. Can the authors explain the choice of the term in the title?

Response: On Page 4, we have clarified the use of “Joint Inflammation model” instead of RA model, based on the published characterization and use of this model and have added 3 references to clarify the differences between the CFA model and RA models: “To further delineate the molecular and cell biological pathways in intestinal responses to joint inflammation, we used a rat model in which local joint inflammation is induced by intra-articular inject

---

## [Decision Letter · Decision Letter 1]

17 Nov 2025

Evidence of an Allostatic Response by Intestinal Tissues Following Induction of Joint Inflammation

PONE-D-25-36861R1

Dear Dr. Moran,

We’re pleased to inform you that your manuscript has been judged scientifically suitable for publication and will be formally accepted for publication once it meets all outstanding technical requirements.

Kind regards,

Andre van Wijnen, PhD

Academic Editor

PLOS ONE

Additional Editor Comments (optional):

Reviewers' comments:

Reviewer's Responses to Questions

**Comments to the Author**

Reviewer #1: All comments have been addressed

Reviewer #2: All comments have been addressed

2. Is the manuscript technically sound, and do the data support the conclusions?

Reviewer #1: Yes

Reviewer #2: Yes

3. Has the statistical analysis been performed appropriately and rigorously?

Reviewer #1: Yes

Reviewer #2: Yes

4. Have the authors made all data underlying the findings in their manuscript fully available?

Reviewer #1: Yes

Reviewer #2: Yes

5. Is the manuscript presented in an intelligible fashion and written in standard English?

Reviewer #1: Yes

Reviewer #2: Yes

Reviewer #1: The manuscript has been improved greatly. I have no further comments, authors have addressed all my comments. Thakns.

Reviewer #2: Dear authors,

I am satisfied with the answers. In its current format, the manuscript can be accepted.

**Do you want your identity to be public for this peer review?** For information about this choice, including consent withdrawal, please see our Privacy Policy

Reviewer #1: No

Reviewer #2: No

---

## [Editor Report · Acceptance letter]

PONE-D-25-36861R1

PLOS One

Dear Dr. Moran,

I'm pleased to inform you that your manuscript has been deemed suitable for publication in PLOS One. Congratulations! Your manuscript is now being handed over to our production team.

Kind regards,

on behalf of

Dr. Andre van Wijnen

Academic Editor

PLOS One